# Lagrangian betweenness as a measure of bottlenecks in dynamical systems with oceanographic examples

Enrico Ser-Giacomi [1✉], Alberto Baudena [2], Vincent Rossi [3], Mick Follows[1], Sophie Clayton[4], Ruggero Vasile[5,6], Cristóbal López [7] & Emilio Hernández-García [7]

The study of connectivity patterns in networks has brought novel insights across diverse fields ranging from neurosciences to epidemic spreading or climate. In this context, betweenness centrality has demonstrated to be a very effective measure to identify nodes that act as focus of congestion, or bottlenecks, in the network. However, there is not a way to define betweenness outside the network framework. By analytically linking dynamical systems and network theory, we provide a trajectory-based formulation of betweenness, called Lagrangian betweenness, as a function of Lyapunov exponents. This extends the concept of betweenness beyond the context of network theory relating hyperbolic points and heteroclinic connections in any dynamical system to the structural bottlenecks of the network associated with it. Using modeled and observational velocity fields, we show that such bottlenecks are present and surprisingly persistent in the oceanic circulation across different spatio-temporal scales and we illustrate the role of these areas in driving fluid transport over vast oceanic regions. Analyzing plankton abundance data from the Kuroshio region of the Pacific Ocean, we find significant spatial correlations between measures of diversity and betweenness, suggesting promise for ecological applications.

[1] Department of Earth, Atmospheric and Planetary Sciences, Massachusetts Institute of Technology, Cambridge, MA, USA. [2] Sorbonne Université,CNRS, Laboratoire d'Océanographie de Villefranche, UMR 7093 LOV, Villefranche-sur-Mer, France, Villefranche-sur-Mer, France. [3] Mediterranean Institute of Oceanography (UM110, UMR 7294), CNRS, Aix Marseille Univ., Univ. Toulon, IRD, Marseille, France. [4] Old Dominion University, Norfolk, VA, USA. [5] UP Transfer GmbH, Potsdam, Germany. [6] GFZ German Research Centre for Geosciences, Potsdam, Germany. [7] IFISC (CSIC-UIB), Instituto de Física Interdisciplinar y Sistemas Complejos, Palma de Mallorca, Spain. ✉email: enrico.sergiacomi@gmail.com

In the last decades, the network formalism has provided new and useful tools in many areas of research. This consists of describing the structural and dynamical features of a system using a set of objects, called nodes, joined by pairwise connections called links[1,2]. Due to the typical complexity of the links' geometry, single nodes can significantly influence the dynamics of large parts of a network[3,4]. Such nodes may correspond not only to the major hubs, i.e., the ones with the largest number of connections, but also to nodes with few links that are, however, crucial to preserve the connectivity of the network[5]. Geometrically, the latter are associated with bottlenecks that constrain and control the connectivity, being an obliged passage to link different pieces of the network (see Fig. 1a).

An explicit measure to assess how much a node behaves as a bottleneck in network theory is the betweenness centrality[1,6]. To calculate betweenness, it is necessary to introduce the concept of paths that are defined as sequences of consecutive links joining pairs of nodes. Betweenness is then obtained by counting the number of paths passing through each node of the network. Depending on the kind of network studied, it can be derived from the whole set of paths, the shortest, the fastest, or the most probable ones[7–11]. Measuring the extent to which a node lies on the existing paths linking other nodes, betweenness has been used to highlight bottlenecks in a variety of different systems, from air transportation networks[12] to the human brain[13].

The bottleneck notion could in principle be generalized beyond the context of network theory. Let us consider indeed a generic dynamical system and describe its dynamics in terms of trajectories portraying the evolution in time and space of an arbitrary set of initial conditions[14,15]. Such approach is often referred to as Lagrangian, in contrast to the Eulerian view where the system is characterized by quantities given at fixed locations. Hence, for a specific interval of time, we can associate to any initial condition at time $t$ and position $\mathbf{x}_0$, a particle following a Lagrangian

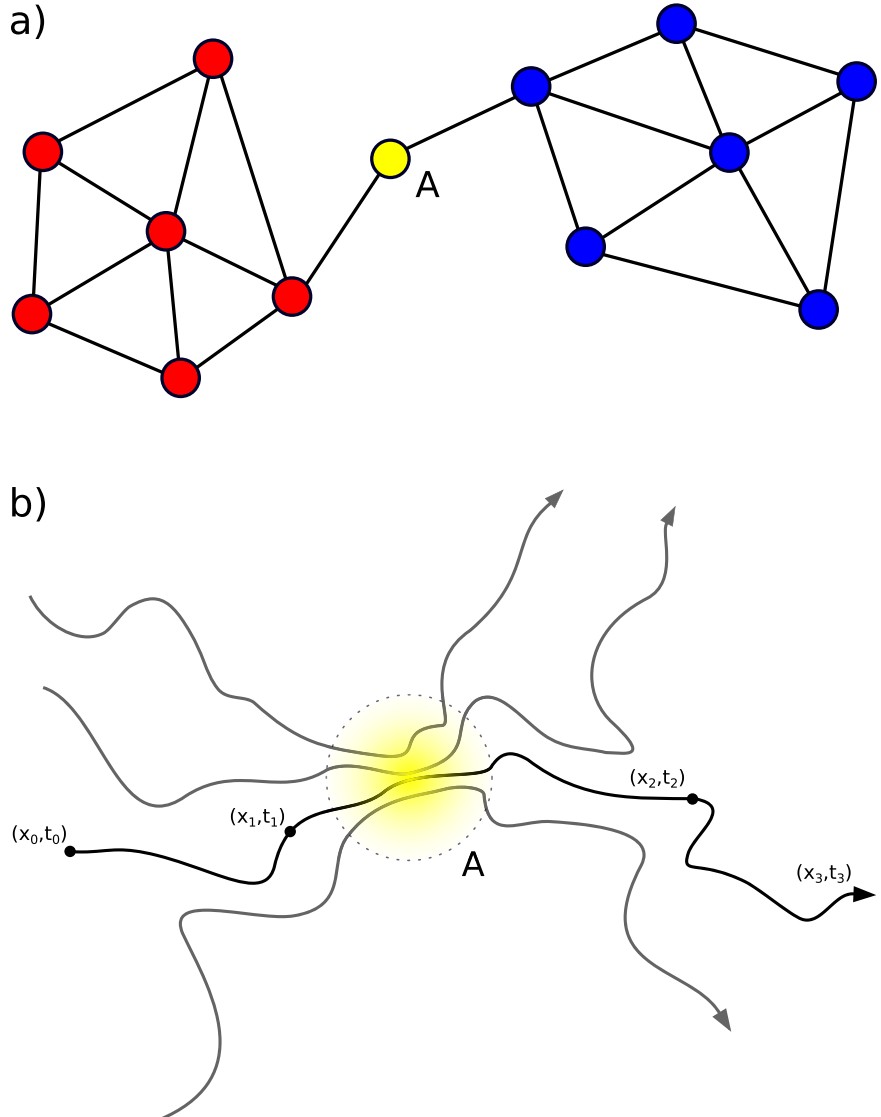

**Fig. 1 Betweenness concept in networks and dynamical systems. a** A small network composed by nodes (colored circles) and links (black segments). The central node A behaves as a bottleneck for the network since all the paths connecting the blue with the red part must go through it. Indeed, node A, despite being attached to only two links, has the maximum value of betweenness centrality. **b** A set of Lagrangian trajectories (black lines) in a generic dynamical system. Each trajectory represents the evolution of an initial condition in space and time. Initial conditions, depending on the case studied, can be associated either to states of a system or to particles and agents moving across it. The yellow region A highlights a region where trajectories from different origins are choked into a narrow passage before being dispersed again. Similarly to node A in **a**, this region would qualitatively represent a bottleneck of the system and should be associated to a high value of betweenness.

trajectory across the system. Trajectories could represent, for example, the paths followed by a system in its state space, or the movement of people in a city, or the dispersion of fluid particles in the ocean or the atmosphere. Bottlenecks can then be related to regions where trajectories converge from disparate origins and are scattered away toward several destinations afterwards (see Fig. 1b). Whatever the system studied, such hotspots are expected to play a crucial role for the maintenance of the system dynamics and for its resilience. In a highly dynamic system such as the ocean, detecting them could have relevant implications for the study, for example, of pollutants spreading[16,17], the dispersion of natural tracers such as biogeochemical species[18], plankton[19–22], propagules or pathogens[23], the spread of invasive species[24], etc.

In dynamical system theory, the term bottleneck has been referred to areas where trajectories spend a long time. These areas are typically found as remnants or ghosts of a just-occurred saddle-node bifurcation[25] or of other types of bifurcations such as the tangent bifurcations leading to intermittency[14]. Also, in the context of chemical reaction dynamics, the concept of bottleneck has been discussed within transition-state theory[26,27]. However, here we aim to a different dynamical property: regions of phase space where many trajectories go through, coming from diverse origins, and then depart also to different destinations (independently of the time spent in the focal region), as directly translated from the concept of betweenness in a network.

To exploit this paradigm, we introduce a mathematical expression for betweenness relying only on the information provided by trajectories sampled across a generic dynamical system. Such quantity, called Lagrangian betweenness, is a function of backward- and forward-in-time finite-time Lyapunov exponents (FTLEs)[14,28–30], which measure the rate of separation of trajectories during a specific time interval. Using an ideal flow system as test-bed, we find a strong resemblance between Lagrangian betweenness patterns derived from trajectories and betweenness calculated using the classical network definition. This allows to uncover an emerging relationship between the concept of bottlenecks in networks and of hyperbolicity[14,31] in dynamical systems. Then we use the Lagrangian betweenness to characterize hidden circulation regimes in realistic geophysical flows, focusing on two paradigmatic oceanic regions: the Adriatic Sea and the Kerguelen area. These two areas are currently the subject of intense research efforts and they represent two examples of very different current systems; a closed basin and an open circulation respectively[32,33]. In both regions, bottlenecks of water transport persistently appear and we show their importance in controlling the exchange of water masses between different areas of the ocean surface. Finally, we compare betweenness fields with measurements of plankton diversity collected during a cruise across the Kuroshio Front, in the Pacific Ocean. We find a remarkable correlation of high-betweenness hotspots with large values of observed plankton diversity, suggesting that fluid transport bottlenecks can also play a significant role in shaping biogeographical patterns in the ocean.

## Results
**Theory**. Betweenness centrality is usually calculated from the number of paths crossing a given node of the network[6,7,10,11]. We provide below an alternative definition of it in terms of products of in- and out-degrees. This allows to use a relation between degrees and Lyapunov exponents[34] to finally derive the formulation of the Lagrangian betweenness.

Let us consider the most general case of a network that is directed, weighted[35] and temporal[36]. Its structure can be entirely encoded in the associated adjacency matrix $\mathbf{A}(t_0, \tau)$ where each element $\mathbf{A}(t_0, \tau)_{ij}$ corresponds to the weight of the link between $i$ and $j$ for a time interval $[t_0, t_0 + \tau]$ (starting at $t_0$ and with a

duration of $\tau$). The basic metric used to quantify how much a single node is connected to the rest of the network is called degree and for directed networks we can distinguish between out- and in-degree[1,2]. Specifically, the out-degree ($K_i^O$) and the in-degree ($K_i^I$) are calculated by counting the number of outgoing and incoming links attached to a given node $i$, respectively

$$K_i^O(t_0, \tau) = \sum_j \begin{cases} 1 & \text{if } \mathbf{A}(t_0, \tau)_{ij} > 0, \\ 0 & \text{otherwise.} \end{cases}$$
$$K_i^I(t_0, \tau) = \sum_j \begin{cases} 1 & \text{if } \mathbf{A}(t_0, \tau)_{ji} > 0, \\ 0 & \text{otherwise.} \end{cases} \quad (1)$$

Without loss of generality, we now take $t_0 = 0$ (this corresponds to considering $[0, \tau]$ as time interval). Then, it is easy to show that the number of two-steps paths crossing the network node $i$ at a generic intermediate time $t$ (with $0 \le t \le \tau$) is the product of the temporal in- and out-degree

$$K_i^I(0, t) \, K_i^O(t, \tau - t). \quad (2)$$

Each step is associated to a precise time interval, $[0, t]$ and $[t, \tau]$ respectively, that matches the interval in which the degrees $K_i^I(0, t)$ and $K_i^O(t, \tau - t)$ are calculated. Our goal is to use Eq. (2) to define a quantity with the meaning of a betweenness measure. Note that the product of degrees in Eq. (2) differs from the classical betweenness centrality formulations in network theory in two aspects: (i) it counts all the paths crossing the node $i$, not just the shortest, fastest, or most probable ones; (ii) it considers only the paths composed of two temporal steps that pass through node $i$ exactly at time $t$. We argue that point (i) is not an issue for our definition, on the contrary, there is an increasing interest in considering centrality measures that take into account the information from all the paths across the network[7,8,9]. Regarding point (ii), to overcome the limitation of forcing the path-crossing to occur exactly and only at time $t$, instead of building paths of arbitrary number of steps and fixed step duration, we look at paths of just two steps of which we can vary the duration in time. This allows the variable $t$, that is the time at which the two steps connect in $i$, to take all the possible values in the interval $[0, \tau]$. Hence, Eq. (2) can be generalized to consider all the two-step paths crossing the node $i$ at any $t$ as follows:

$$\frac{1}{\tau} \int_0^\tau K_i^I(0, t) \, K_i^O(t, \tau - t) dt. \quad (3)$$

Equation (3) represents thus a candidate for a novel continuous-in-time definition of betweenness centrality for any network where the time duration associated to each link can be explored. An analogous definition of betweenness can also be derived for time-independent networks and/or networks with fixed link-duration by using $k$-neighbor degrees ("Methods"). 

To extend our formulation beyond network theory, we need a bridge with the underlying dynamical system of which the network is a representation, as sketched in Fig. 1. We assume that nodes are associated to specific regions of the space in which such system is defined and links symbolizes connections between nodes realized by trajectories. Specifically, we look for a relationship among the degrees of Eq. (3) and the geometry of Lagrangian trajectories in the surrounding of the position corresponding to the node $i$. To this aim, we can use a relation between in/out-degrees and backward/forward-in-time stretching factors ("Methods") derived in the context of flow systems within the Lagrangian Flow Networks (LFNs) framework[34,37]. Stretching factors measure the rate of separation of trajectories in a given time interval and are expressed as exponential functions of the FTLEs[29]. If we take the limit of sufficiently small nodes, we find

the following relations:

$$K_i^O(t_0, \tau) \approx e^{\tau\lambda(\mathbf{x}_i, t_0, \tau)},$$
$$K_i^I(t_0, \tau) \approx e^{\tau\lambda(\mathbf{x}_i, t_0 + \tau, -\tau)}, \qquad (4)$$

where $\lambda(\mathbf{x}_i, t_0, \tau)$ is the standard FTLE computed for a time $\tau$, starting at time $t_0$, at location $\mathbf{x}_i$ (which denotes the position of node $i$).

The last step is to combine Eqs. (3) and (4) to link the betweenness centrality of the network with the FTLEs of the underlying system. In such way, we finally define the Lagrangian betweenness of node $i$ as:

$$B_i^L(0, \tau) = \frac{1}{\tau} \int_0^\tau e^{t\lambda(\mathbf{x}_i, t, -t)} \, e^{(\tau-t)\lambda(\mathbf{x}_i, t, \tau-t)} \, dt. \qquad (5)$$

The integrand in Eq. (5) corresponds to a product of forward and backward stretching factors associated to $\mathbf{x}_i$ at time $t$ and, consistently, $B^L$ is dimensionless. Note that, since relative stretching measures such as $\lambda$ remain invariant under coordinate transformation, $B^L$ is frame-invariant too. Under some stretching regimes, it is possible to solve analytically the integral of Eq. (5) to obtain analytical approximations for $B^L$ ("Methods"). However, numerical evaluations of Eq. (5) can be easily obtained through a time discretization ("Methods"). Then, $B^L$ can be straightforwardly compared with a betweenness measure explicitly calculated from the network theory definition ("Results" and "Methods").

After defining $B_i^L$ in terms of FTLEs, we provide its interpretation from a dynamical systems theory perspective. From Eq. (5), we see that nodes presenting, on average, high values of both backward and forward FTLEs during the interval $[0, \tau]$ are characterized by high $B^L$ (see Fig. 1). Interestingly, in dynamical systems, large values of forward or backward FTLEs highlight the locations of strongly repelling or attracting material surfaces, related to stable or unstable manifolds, respectively[29,38]. Considering the case of two-dimensional motion, their intersections define, at each instant of time, hyperbolic points with eventual heteroclinic and homoclinic connections among them[14,31] (see Fig. 2). In time-dependent systems, such objects move in space spanning hyperbolic trajectories (points) or areas (connections) making their detection more difficult. Indeed, different approaches have been devoted to the tracking of moving hyperbolic points in dynamical systems[31,39]. Now, thanks to Eq. (5), an explicit correspondence emerges between hyperbolic points, heteroclinic/homoclinic connections, and the main bottlenecks of the system. In this sense, $B^L$ provides a clear indication of the role of these features in organizing, limiting, and eventually controlling any trajectory-mediated connectivity and transport processes across a dynamical system. We stress that there is a crucial distinction between hyperbolic points and connections in terms of transport, while relative velocities of trajectories in the neighborhood of hyperbolic points are close to zero, velocities along connections can be significantly large[29]. In fact, bottlenecks are not determined locally by the magnitude of fluxes but rather by the entire topology of the system that amalgamate around them trajectories coming from diverse origins and going to several other destinations[1,10].

Though many kinds of systems can be studied with the paradigm proposed in the previous paragraphs, in the following applications, we concentrate on flow systems: first on a theoretical flow, then on the transport patterns of two different oceanic regions and finally on the relationships between ocean circulation and plankton diversity.

### Testing Lagrangian betweenness in a theoretical model.
In this section, we calculate the Lagrangian betweenness in a two-dimensional, theoretical flow system, called double-gyre[29]; a well-known benchmark for study mixing and transport in fluid

dynamics ("Methods"). The flow is composed of two gyres side by side that rotate in opposite directions while the vertical boundary between them oscillates periodically in time generating complex trajectories and patterns (Fig. 3).

In Fig. 4, we show that the $B^L$ field evaluated numerically from Eq. (20): high values of Lagrangian betweenness are found close to the boundaries of the system and in a narrow semicircular pattern splitting the domain vertically. Intermediate values are mainly found in a rectangular band centered on the midline of the domain. Its width matches the region spanned by the boundary separating the two gyres in the flow.

While high values of $B^L$ at the borders are clearly due to boundary effects, the semicircular line in the middle of the domain needs further analysis to be properly understood. To this aim, in Fig. 5, we plot snapshots of the difference between the exponents of the two factors inside the integral of Eq. (5), i.e., $\lambda(\mathbf{x}_i, t, \tau - t) - \lambda(\mathbf{x}_i, t, - t)$, for $t = 5, 6, 7, 8, 9, 10$ and keeping $\tau = 15$. Hence, red and blue regions present high values of forward and backward FTLE, respectively, and can be ultimately related to stable and unstable manifolds of the system. Consequently, the crossing point of backward- and forward-in-time FTLE ridges identifies the instantaneous position of a moving hyperbolic point (like the one of Fig. 2). Strikingly, the trajectory of such point matches perfectly the ridge of high $B^L$ confirming that hyperbolic regions are associated with high values of betweenness.

Furthermore, we explicitly prove such relationship comparing the $B^L$ field with the betweenness centrality obtained using the standard network definition. First, we build a set of networks, the so-called LFNs[34], describing fluid transport in the double-gyre system using the same parameters setup ("Methods"). Then, we calculate betweenness for each network node matching the time scales used for the Lagrangian betweenness calculation. Note that, a direct comparison with Fig. 4 is not possible due to the intrinsic

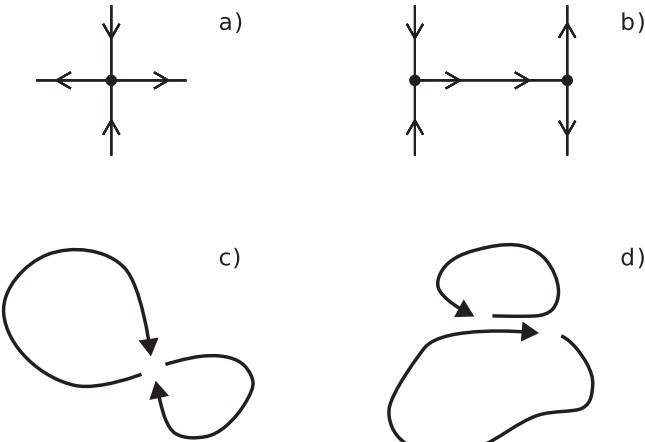

**Fig. 2 Schematic representations of a hyperbolic point and a heteroclinic connection together with associated circulation patterns.** Lines with converging/diverging arrows denote stable/unstable manifolds of the hyperbolic points (represented as black dots), respectively. Note that in (**b**) the manifold that realizes the connection is unstable for the left-hand side hyperbolic point and stable for the right-hand side one. Due to time-dependence, which is weak but present in geophysical flows, patterns like **a** or **b** are weakly perturbed and transformed in so-called moving hyperbolic points and connections. In particular, **c** and **d** sketch two examples of circulation patterns that can be often found in the ocean, respectively, associated to a hyperbolic point and to a heteroclinic connection. They exemplify two gyres sharing a common point or segment of their boundaries that can be associated to the elementary sketches (**a**) and (**b**) and are therefore expected to display high Lagrangian betweenness values.

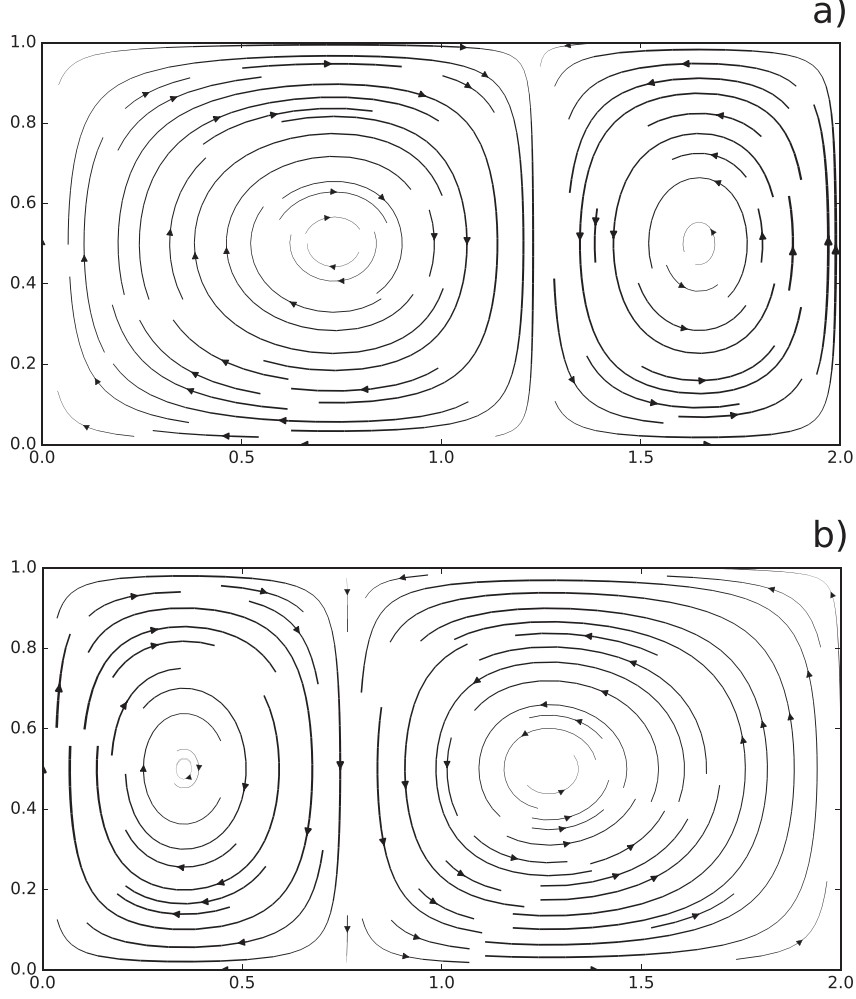

**Fig. 3 Streamlines of the double-gyre flow.** Plots of streamlines for $t = 2.5$ (**a**) and $t = 7.5$ (**b**), thicker lines are associated to higher velocities. The circulation is dominated by two gyres rotating in opposite directions separated by a vertical boundary that oscillate periodically along the horizontal direction. This time-dependence generates chaotic trajectories that induce complex mixing patterns across the entire flow domain.

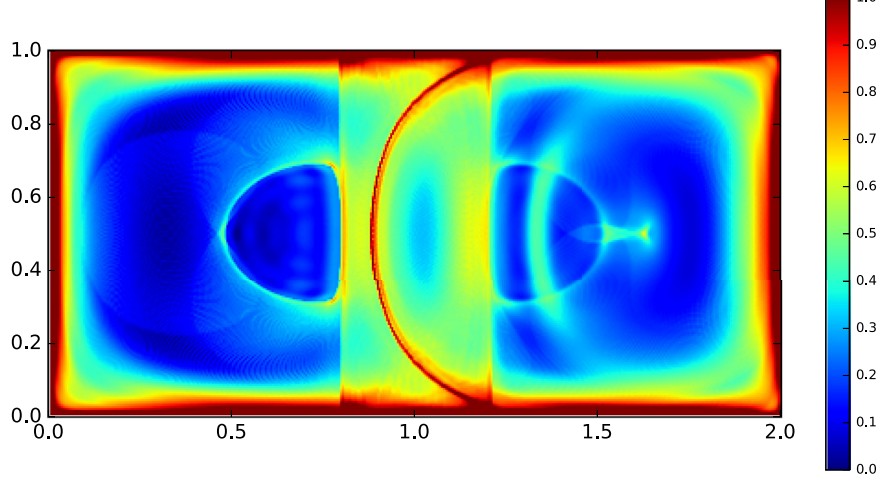

**Fig. 4 Plot of Lagrangian betweenness for the double-gyre flow with a normalized logarithmic color map.** Here, we performed a fine numerical integration (with $N = 300$) to converge to the analytical expression for Lagrangian betweenness of Eq. (5). Higher values of Lagrangian betweenness are found at the boundaries and across the narrow semicircular line splitting the domain vertically. The region spanned by the moving line separating both gyres is clearly highlighted by intermediate values of betweenness.

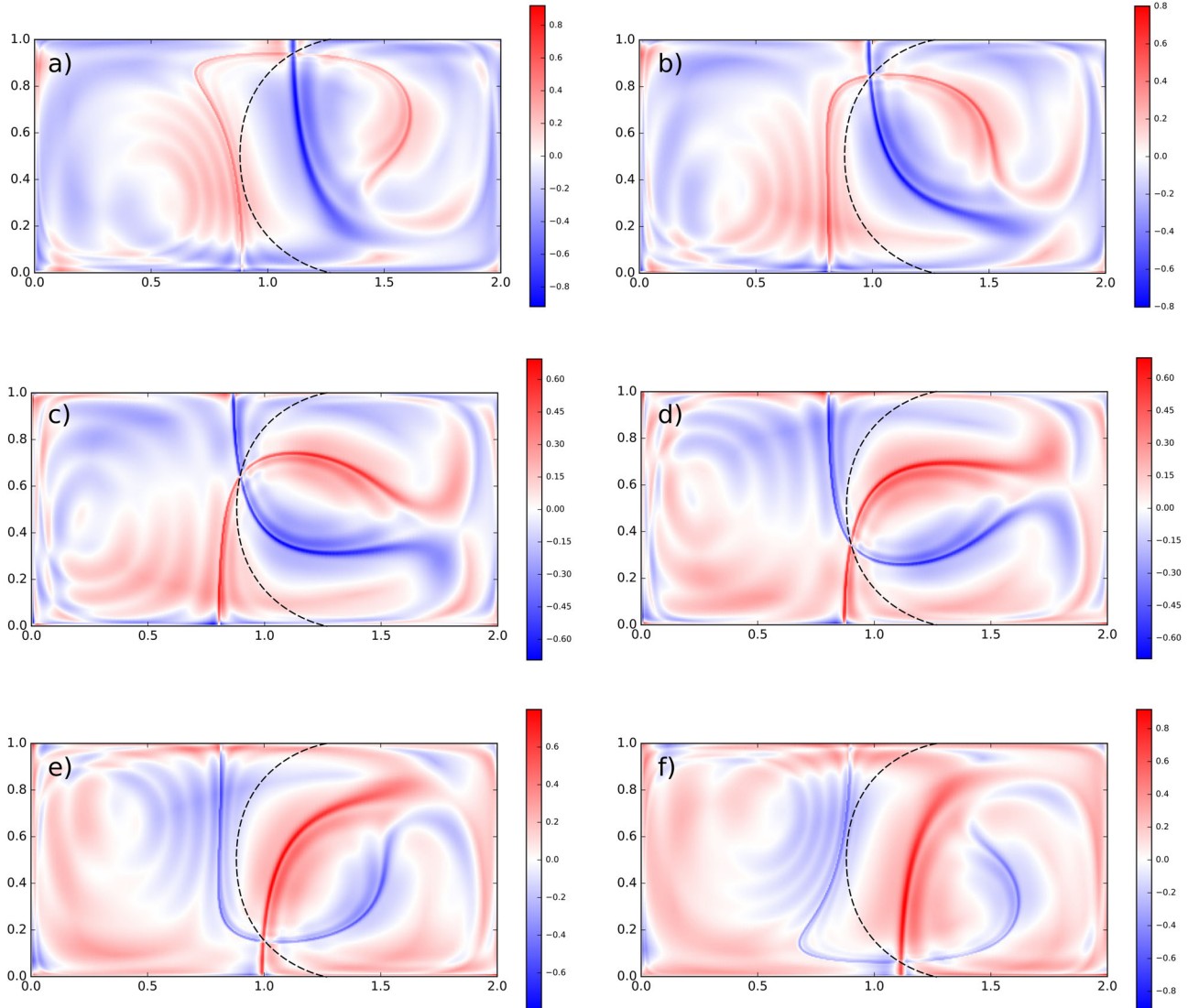

**Fig. 5 Finite-time Lyapunov exponents (FTLEs) fields in the doube-gyre. a–f** Superimposition (by difference) of forward (red) and backward (blue) FTLEs fields for the double-gyre flow at different intermediate times ($t = 5, 6, 7, 8, 9, 10$) and matching the time interval [0, 15]. The dashed black line is the $B^L$ ridge of Fig. 4. We clearly see the moving location of a hyperbolic trajectory, detected by the intersection of both main ridges in backward and forward FTLE, changing its position for different intermediate times. Such hyperbolic trajectory matches perfectly the $B^L$ ridge.

discrete nature of the standard network betweenness, and we can thus only test on few temporal steps. Moreover, since the network betweenness calculation is much more numerically demanding, we compare the patterns of the two measures on a coarser resolution grid. As results, we obtain a strong resemblance of the spatial patterns between the two measures confirmed by large Spearman correlation coefficients ("Methods," Supplementary Fig. 1).

**Example applications to fluid transport in the ocean**. To apply our framework in a realistic geophysical setting, we exploit velocity fields of the ocean as modeled by a high-resolution hydrodynamic model and as measured by satellite altimetry ("Methods"). We focus on the month of December as representative of the the typical winter/summer conditions in the northern/southern hemisphere while we select randomly two different years.

Our first application focus on the Adriatic Sea that is a relatively closed sub-basin of the Mediterranean Sea whose surface circulation is dominated by two large wind-driven gyres[32,40]. On top of this basic structure, a strong sub/mesoscale variability superimposes its dynamical signatures. This complex surface circulation as well as

the intense human and biological activity in this region have stimulated strong research interests in the last years[41].

We plot in Fig. 6 (top) the $B^L$ field computed from the high-resolution-simulated velocity fields in the Adriatic Sea, for the 1st of December 2013 and with an integration time $\tau$ of 15 days. It identifies a small circular area with very large values of $B^L$ (almost one order of magnitude greater than the surrounding) located southeast of the Pelagosa Islands. Computing the average sea-surface height (SSH) on the same period (Supplementary Fig. 2), we note that the region is characterized by two cyclonic gyres that present a contact point in the same approximate location of the "Pelagosa peak" of $B^L$, reminiscent of the hyperbolic geometry of Fig. 2c.

In order to quantify explicitly the influence of the Pelagosa $B^L$ peak on the surrounding circulation, we fill the interior of the two gyres with tagged Lagrangian particles and we simulate their trajectories in between the 1st and the 15th of December. To draw the boundary between the gyres interior and the exterior, we set a SSH threshold of −20 cm and we seed particles only for SSH values smaller than the threshold, associated thus to the core of both gyres (see Supplementary Fig. 2). In Fig. 6 (bottom, four panels), we also

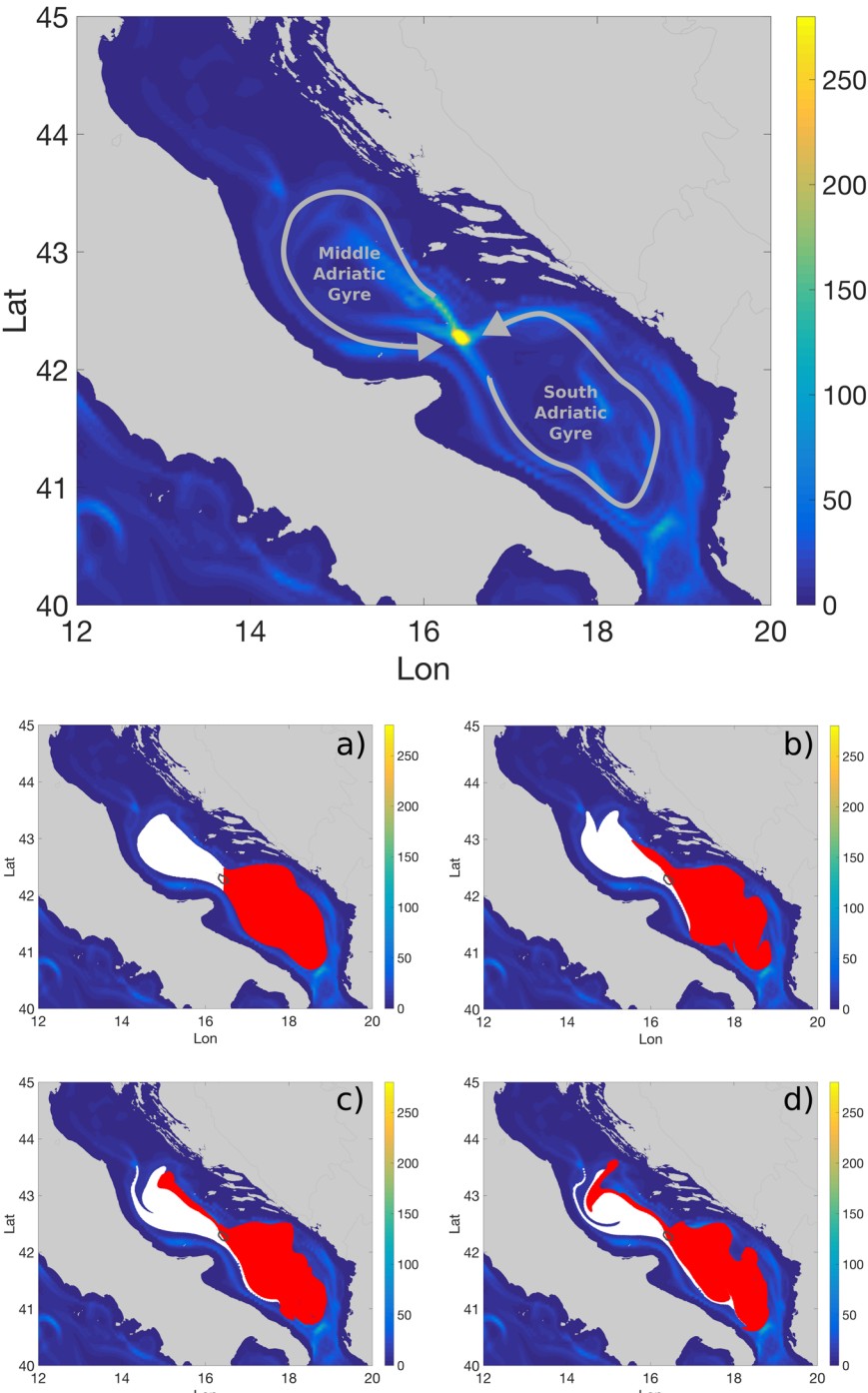

**Fig. 6 Lagrangian betwenness in the Adriatic Sea.** On the top: $B^L$ field calculated for the 1st of December 2013 and with an integration time $\tau$ of 15 days from model data of the Adriatic Sea. The gray arrows sketch the circulation pattern: two regional cyclonic gyres sharing a contact point exactly where the "Pelagosa peak" of $B^L$ is located. On the bottom, **a**–**d** evolution of particle patches seeded in the interior of the gyres for different intermediate times (1st, 5th, 10th, and 15th of December) from model data. The white patch occupies initially the Middle Adriatic Gyre and the red one the South Adriatic Gyre while the contour of the $B^L$ peak is denoted by a solid gray line. Note that exchanged water between the two gyres always flows in a small region around the $B^L$ peak (see also the Supplementary Video 1 for the full animation).

show the evolution of the aforementioned particle patches at different intermediate times (Supplementary Video 1). The patch evolution confirms that the Pelagosa peak is associated to the presence of a surprisingly stable and steady hyperbolic point, crossed by a stable manifold in the east/west direction and by a unstable one in the north-west/south-east direction.

Moreover, looking at the water origins in the interior of the Pelagosa peak, we find that such area corresponds to the only

place in the basin where it is possible to encounter southern water advected toward the northern gyre and, at the same time, northern water transported to the southern gyre. This means that, similarly to the yellow circle of Fig. 1b, the Pelagosa peak correctly exemplifies what an oceanic bottleneck represents: a very small portion of the ocean surface that permits the exchange and subsequent mixing of two water masses, which occupied two, otherwise disconnected, larger oceanic regions.

To prove the robustness of this structure to velocity fields of different origin and resolutions, we repeat the $B^L$ calculation varying the integration time $\tau$ and using also an altimetry-derived dataset ("Methods"). We find a remarkable regularity in the position of the peak and a consistent increase of its absolute value with $\tau$, both for model and satellite observations (Supplementary Fig. 3). We also perform a temporal average of the $B^L$ field from the regional altimetry-derived velocity field across the years 2002–2013 (Supplementary Fig. 4). Again, the Pelagosa peak is clearly distinguishable confirming a striking regularity of this pattern also from simulations based on satellite observations and across several years.

Finally, we have checked that the modulus of the velocity field does not present any persistent stagnation point in the same region, neither in December 2013 nor when averaged across different years. This suggests that despite the absence of an Eulerian saddle point, high-betweenness hotspots emerge driven by a purely Lagrangian dynamics.

Our second application is on the Kerguelen region that is located in the Indian sector of the Southern Ocean and it is characterized by the interaction of the energetic Antarctic Circumpolar Current with a complex topography[33]. It constitutes also one of the ten largest marine protected areas in the world and understanding its circulation patterns is significant for regional ecology and marine biology[42].

Using observed altimetry fields to compute trajectories, we show in Fig. 7 (top) the $B^L$ field in the region north-east of the Kerguelen Islands for the 1st of December 2007 with integration time $\tau = 20$. We can clearly identify an elongated high-betweenness strip centered around 47.7S 75.0E with an approximate length of 200 km and a width of 25 km, we delineate it as the locations having $B^L > 100$. Its location likely coincides with the place where the Polar Front meets the fast eastward flowing Antarctic Circumpolar Current delimited by the Subantarctic Front[33].

In Fig. 7 (bottom, four panels), we also plot the evolution of two tagged patches of water with different origins that flow across the high-betweenness strip at different intermediate times between the 1st and the 20th of December 2007 (Supplementary Video 2). The two patches together represent all the surface water particles that, in the time window considered, touch a point with a betweenness value equal or greater than the threshold used to delimit the strip.

To distinguish to which patch each Lagrangian particle is belonging to, we apply a threshold on the backward-in-time drift distances for each of them computed for the 7 previous days (Supplementary Fig. 5). We assign the particles with larger drift to the Circumpolar Current patch and the ones with short drift to the Polar Front patch. This choice is supported by the fact that, in the Kerguelen region, the flow in the Polar Front is weaker and more meandering than the Circumpolar Current and this is also reflected in the strong bimodality of the drift distribution.

The "hourglass" shape formed by these two Lagrangian patches (Fig. 7) indicates an underlying mean circulation resembling the one sketched in Fig. 2b, d, suggesting thus the presence of a heteroclinic-like structure. Consistently with this interpretation, three fundamental features characterize the particle's evolution close to the high $B^L$ area: (i) a strong convergence toward the strip of the particles initialized in the west, equivalent to a backward-in-time dispersion for particles entering in the strip from the northwestern edge, (ii) a similarly strong forward-in-time dispersion for particle exiting from the southeastern edge, and (iii) a rapid and coherent southeasterly flow along the entire strip.

Consequently, the tracer separation distance in the transversal direction of the main eastward flow is of the order of 250 km prior and after being funneled into a mere 25 km wide strip.

Similar to the Adriatic Sea example, the high-betweenness strip represents thus a tiny oceanic region that sees different water masses converging together and then rapidly spreading away after being partially mixed. Such dense congestion of trajectories illustrates perfectly the concept of bottleneck sketched in Fig. 1: a narrow passage in the ocean surface that sees water particles coming from disparate origins and going to many different destinations.

Finally, similarly to the Adriatic case, we confirm the robustness of the pattern detecting the high $B^L$ area in the same location also in maps calculated with different values of $\tau$ (Supplementary Fig. 6).

**Plankton diversity and Lagrangian betweenness in the Kuroshio Front**. Lagrangian betweenness can also provide a framework for interpreting microbial community structure in fluid environments. We hypothesize that regions of high Lagrangian betweenness promote biodiversity by the confluence of upstream heterogeneous populations. Locally, fine-scale dispersal processes intermingle them and subsequent divergence spreads the newly mixed population widely, seeding downstream environments.

Here, we focus on the Kuroshio Extension Front, a highly dynamical region where the confluence of energetic currents with different propieties has a strong influence on planktonic community composition[20]. We examine the relationship of betweenness and diversity patterns in two independent datasets collected during a cruise in October 2009; microscopy-based abundances of plankton types and phylogenetically derived abundances of *Ostreococcus* clades[43,44]. To locally quantify plankton diversity, we calculate the species evenness[45] at each sampling site in both datasets ("Methods"). At the same time, we use velocities from a global, high-resolution ocean reanalysis product to compute Lagrangian betweenness fields during the sampling period ("Methods").

In Figure 8 (top panel), we show the cruise track and the betweenness field on 20 October 2009 with $\tau = 7$ days. Two high-betweenness strips are apparent and reflect the dominant eastward flow associated within the Kuroshio Extension Front. To allow a direct comparison with the estimated plankton diversity, we associate to each sampling location a value of betweenness equal to the average of the field in a circle of 0.2° of radius around the site (approximately twice the spacing between samples), calculated on the exact sampling time. We find a remarkable, positive and significant correlation of evenness with betweenness in both datasets, as illustrated in Fig. 8 (bottom panels), which show the results from both the microscopy abundances (left) and from *Ostreococcus* clades abundances (right) for each sampling site. While being just a single, limited and opportunistic analysis, this suggests that further investigation of ecological bottlenecks, as revealed by betweenness, will be valuable.

## Discussion

Dynamical systems theory and network theory have been successfully used to characterize the structure and the dynamics of a variety of natural and human complex systems. However, only few theoretical connections between them exists[1,14]. Here, we contribute to bridge this gap by introducing the concept of Lagrangian betweenness. On the one hand, this allows indeed the quantitative identification of bottlenecks of trajectories in dynamical systems and reveals their function in terms of transport, mixing, and connectivity. On the other hand, from the network side, Eq. (5) provides a interpretation of betweenness linking it to hyperbolic and heteroclinic dynamics[14,15].

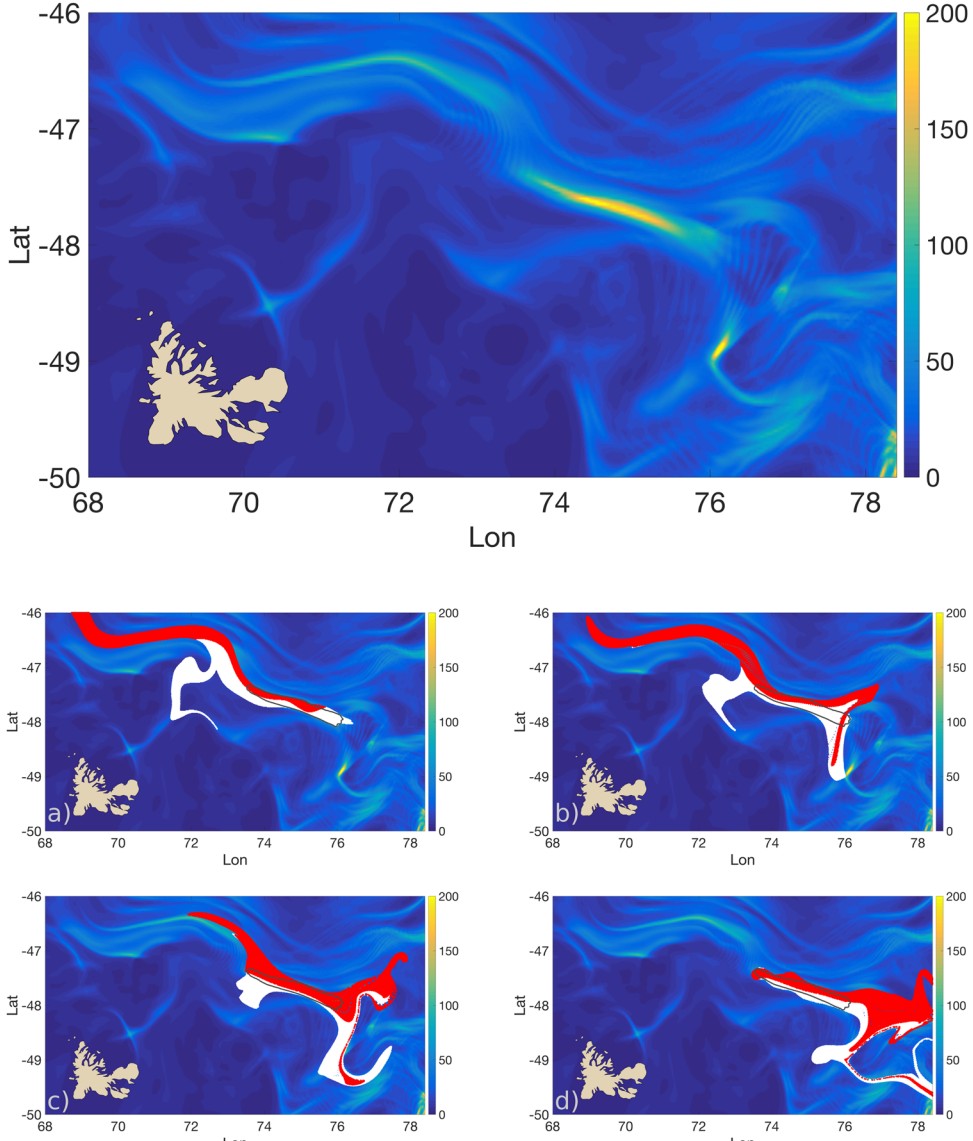

**Fig. 7 Lagrangian betwenness in the Kerguelen region.** On the top: $B^L$ field from altimetry data in the region north-east of the Kerguelen Islands for the 1st of December 2007 with integration time $\tau = 20$. A marked high-betweenness strip is located in the area where the Polar Front meets the Antarctic Circumpolar Current. On the bottom, **a**–**d** Evolution of the Circumpolar Current patch (red) and the Polar Front patch (white) flowing across the high-betweenness strip (delimited by the solid gray contour) at different intermediate times (1st, 7th, 14th, and 20th of December) from altimetry data. Waters from different current systems are funneled through the 20 km wide strip being partially mixed; they separate and disperse shortly after leaving this high-betweeness strip (see also the Supplementary Video 2 for the full animation).

Our definition of betweenness accounts for all the paths crossing a given node, not just a subset of them (e.g., most probable, fastest, shortest ones)[7–10]. Indeed, such approach seems to be the most natural when there is no possibility for the transported quantity to actively choose the most convenient pathways[7,8]. This is the case of any system where trajectories are driven by external forces not involved in the routing process, including, of course, fluid flows.

Moreover, the continuous-in-time definition that we propose allows us to obtain a betweenness measure directly from degrees without passing through the definition of paths. Notably, also in networks where the link duration is fixed, there is still the possibility of using a formulation similar to Eq. (3) in which the degrees are replaced by $k$-neighbor degrees (see "Methods"). This poses the question if betweenness centrality is an intrinsic property of a network or a spandrel that is implicitly determined by the degree distribution, similarly, for instance, to the recently demonstrated case of nestedness[46].

The possibility of obtaining a betweenness measure directly from trajectories without passing through the process of network construction and analysis constitutes an important advantage. Indeed, particularly for temporal networks, this provides a massive decrease of computational time as well as prevent possible sensitivity issue related to, for instance, time discretization and weights thresholding[10,47].

We also stress that the applications presented here are restricted to two-dimensional systems while, in principle, our approach can be applied to higher-dimensional cases. However, such kind of extensions could present significant challenges due to the presence of multiple directions of maximal expansion and compression. Future studies should address whether high-betweenness patterns are still present and relevant in such dynamical conditions.

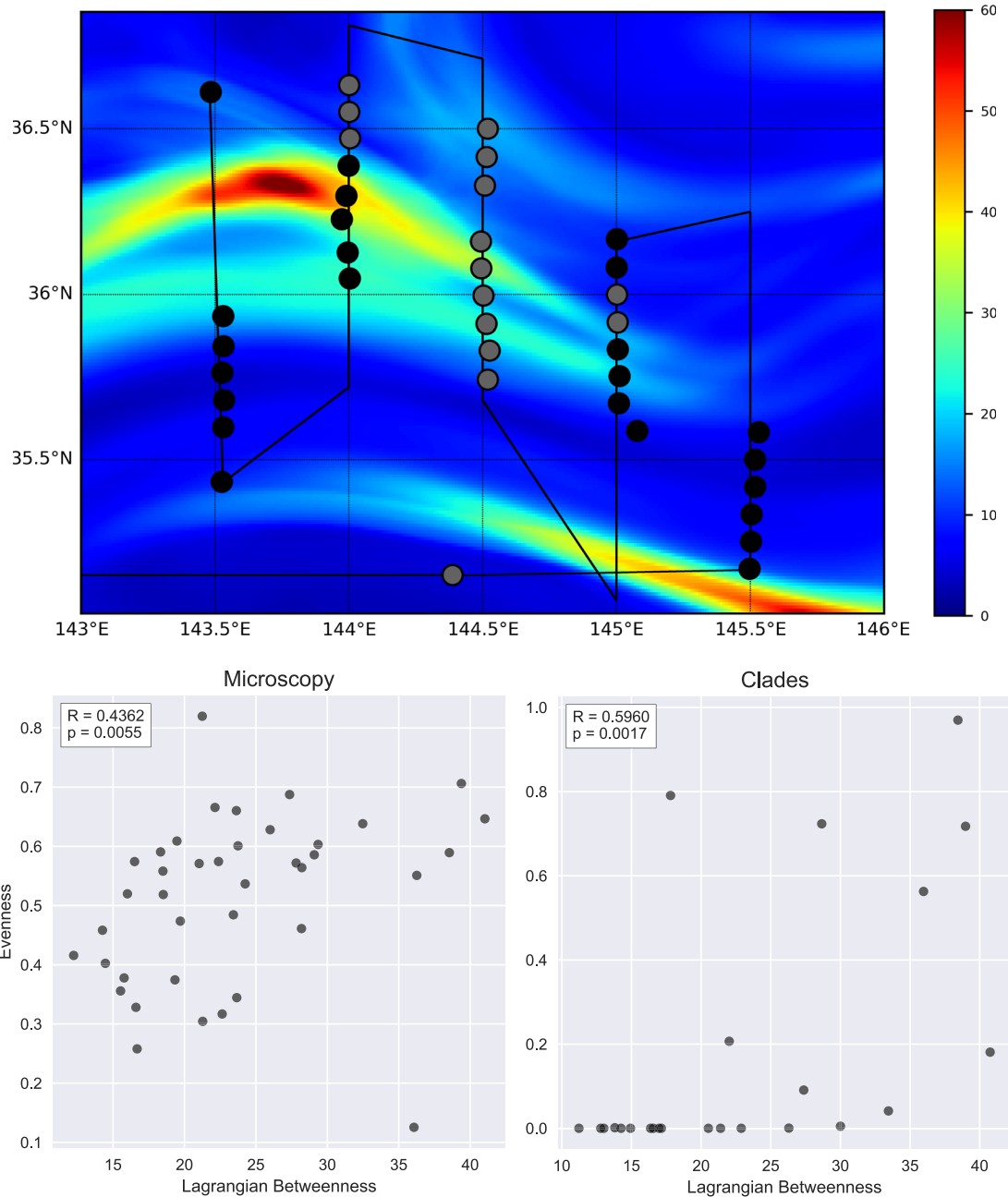

**Fig. 8 Lagrangian betweenness and plankton diversity.** On the top: $B^L$ field calculated for the 20th of Octuber 2009 with an integration time $\tau$ of 7 days from model data of the Kuroshio Front. The black solid line is the cruise track while the dots correspond to different sampling sites. The color of the dots represents the data availability at the sampling location, black when both microscopy and Otreococcus clades data are collected and gray when only microscopy is available. On the bottom: scatter plot of $B^L$ versus species evenness calculated from microscopy (left panel) and clades (right panel). Each dot is associated with a sampling site. Evenness is calculated as a normalized Shannon entropy from relative abundances. To each site, a betweenness value equal to the average of the field in a circle of 0.2° of radius around the site is assigned, calculated on the exact sampling time. Spearman correlation coefficients ($R$) and $p$ values (p) are shown in the top-left inset of each scatter plot indicating a significant and positive correlation of evenness with betweenness.

All in all, if betweenness has proved to be a fundamental measure to assess locally the vulnerability of complex networks, these properties can also be linked now with the concepts of predictability and chaos associated to the underlying system dynamics. Indeed, several studies associated high-betweenness nodes to the more vulnerable parts of a network to local disturbances or attacks[48] while, on the other hand, large values of Lyapunov exponents highlight the most chaotic and less predictable regions in a

dynamical system[14]. Thus, the connection of this network concept with well-characterized behaviors in dynamical systems, such as divergence of trajectories or controllability[49], could open promising avenues of research.

Even though a wide variety of structures, including some resembling those revealed by Lagrangian betweenness, have been already detected in laboratory and in geophysical flows[19,39,40,50], they have not been explicitly associated with transport

bottlenecks. Here we show that high $B^L$ areas identify fluid masses coming from several origins and going to many other destinations, with the difference that, for hyperbolic points (Adriatic Sea), the fluid velocities are typically modest, while for heteroclinic connections (Kerguelen), are larger.

Surprisingly, in our first two oceanic examples, we also find that such points or connections are much more stable and persistent than expected, despite the time variability of realistic flows that, evidently, is not sufficiently strong to completely disrupt them[31,39]. The robustness of such structures is confirmed by a series of considerations: (i) they are detectable both from high-resolution models and from SSH measurements, (ii) they result to be robust for different $\tau$'s and different resolutions of the initialization grid, (iii) they can be found across several years close to the same position, possibly constrained by the bathymetry. Moreover, since both LFNs diagnostics and Lyapunov exponents were proven to be robust to various input parameters and flow fields[51,52], we expect that Lagrangian betweenness will also share the same reliability across a wide range of oceanographic applications.

Since high Lagrangian betweenness regions represent the optimal compromise between the number of water origins and destinations of potentially different hydrodynamical and biogeochemical properties, we tested the hypothesis that they could play a role in shaping diversity patterns of marine ecosystems[19,22,53,54]. In our Kuroshio Front application, despite the use of a reanalysis product not adapted to the region for the trajectory calculation, we find promising statistical correlations between Lagrangian betweenness fields and local plankton diversity measured from two independent observational datasets. However, the large-scale influence of high-betweenness hotspots on ecosystem functioning should be further investigated in detail. Indeed, following a water parcel converging toward a high-betweenness area, the ecological community associated to it would initially experience an increase of diversity due to the exchanges with other parcels coming from different biogeographical regions[55]. But afterwards, different ecological conditions could either favor neutrality or competitive exclusion, thus maintaining or reducing such initial high diversity. In both cases, the effect of such changes will be rapidly dispersed across large oceanic regions[22,56].

From a more applied perspective, the effectiveness of betweenness in assessing systems sensitivity could also contribute to the design of optimized observing and monitoring systems on targeted areas of the ocean[57,58]. High-betweenness hotspots could indeed provide early-warning signals to anticipate when and where the marine environment would be affected by multiple stressors such as pollutants, pathogens, invasive species, or climatic-induced threats.

Beyond the oceanic focus, Lagrangian betweenness could be readily used in other contexts. It could detect, for instance, dispersal hubs for propagules, pollutants, or pathogens in atmospheric and urban flows[59–61], help in the study of the predictability of the climate systems[62,63] and provide insights on turbulent transport in astrophysical fluids[64].

## Methods

**$k$-neighbor degrees and the time-independent case.** For the case of time-independent networks, we lose the temporal dimension and the degrees will not depend on time, but we still have the information of the number of steps needed to build a given path across the network. In this sense, long-range connections will be realized across a larger number of steps than the shorter ones. We denote the weighted, time-independent adjacency matrix of a given time-independent network as $\mathbf{A}$. We also define the correspondent unweighted, time-independent adjacency matrix as:

$$\mathbf{U}_{ij} = \begin{cases} 1 & \text{if } \mathbf{A}_{ji} > 0 \\ 0 & \text{otherwise} \end{cases} \tag{6}$$

We introduce the time-independent $k$-neighbor out- and in-degrees as:

$$\mathcal{K}_i^{O^{(k)}} = \sum_{j_1, j_2, \dots, j_k} \mathbf{U}_{ij_1} \mathbf{U}_{j_1 j_2} \dots \mathbf{U}_{j_{k-1} j_k} ,$$
$$\mathcal{K}_i^{I^{(k)}} = \sum_{j_1, j_2, \dots, j_k} \mathbf{U}_{j_1 i} \mathbf{U}_{j_2 j_1} \dots \mathbf{U}_{j_k j_{k-1}}, \tag{7}$$

where we set $\mathcal{K}_i^{O^{(0)}} = \mathcal{K}_i^{I^{(0)}} = 1$. Following the approach presented in "Results" and using Eq. (7), we finally find an analogous expression to Eq. (3) for the time-independent case

$$\frac{1}{k} \sum_{l=0}^{k} \mathcal{K}_i^{I^{(l)}} \mathcal{K}_i^{O^{(k-l)}} . \tag{8}$$

**FTLEs and stretching factors.** In dynamical system theory, a quantity to characterize locally dispersion and mixing is the FTLE[14]. The FTLE, computed during a duration $T$ for a trajectory that started at time $t$ at initial location $\mathbf{x}_0$, is defined as:

$$\lambda(\mathbf{x}_0, t; T) = \frac{1}{2|T|} \log |\Lambda_{max}|, \tag{9}$$

with $\Lambda_{max}$ the largest eigenvalue of the right Cauchy–Green deformation tensor $\mathcal{C}(\mathbf{x}, t; T)$ defined as[28,29,62]:

$$\mathcal{C}(\mathbf{x}, t; T) = \frac{d\phi_t^{t+T}(\mathbf{x})^*}{d\mathbf{x}} \frac{d\phi_t^{t+T}(\mathbf{x})}{d\mathbf{x}} \tag{10}$$

where $\phi_t^{t+T}(\mathbf{x})$ is the so-called flow map that gives the position of the trajectory started at $\mathbf{x}$ after a time $T$. The matrix $\frac{d\phi_t^{t+T}(\mathbf{x})}{d\mathbf{x}}$ is the Jacobian (derivatives with respect to the initial condition) of the flow map and $\frac{d\phi_t^{t+T}(\mathbf{x})^*}{d\mathbf{x}}$ is its adjoint. Equation (9) can be expressed also as:

$$\lambda(\mathbf{x}_0, t; T) = \frac{1}{|T|} \log \left( \frac{||\delta\mathbf{x}_0(t, T)||_{max}}{||\delta\bar{\mathbf{x}}_0(t)||} \right), \tag{11}$$

where $||\delta\bar{\mathbf{x}}_0(t)||$ is the initial separation between infinitesimally close initial conditions located around $\mathbf{x}_0$ at time $t$ and aligned with the eigenvector of $\Lambda_{max}$; while $||\delta\mathbf{x}_0(t, T)||$ is the final separation of those particles at time $t + T$, being the maximum possible separation resulting from all the directions of particle separation $\delta\mathbf{x}_0(t)$. Note that, from Eq. (10), we have that the square root of the eigenvalue $\Lambda_{max}$ corresponds also to the singular value of the Jacobian matrix of the flow map from which the Cauchy–Green tensor is built[28,29,62]. FTLEs characterize thus the maximum logarithmic separation rate, over an interval of time $T$, around $\mathbf{x}_0$; for $T > 0$ and $T < 0$, we obtain the forward- and backward-in-time FTLE, respectively. Hence, an initial sphere or circle of diameter $d$ located in $\mathbf{x}_0$ at time $t$ would be elongated at time $t + T$ by a stretching factor $s(\mathbf{x}_0, t; T)$ defined as:

$$s(\mathbf{x}_0, t; T) = e^{T\lambda(\mathbf{x}_0, t; T)}. \tag{12}$$

Practically, for a given system, FTLEs are obtained from Lagrangian trajectories of a set of initial conditions during a fixed time interval. Such trajectories are usually reconstructed using modeled or observed gridded velocity fields, or real trajectories from Lagrangian tracers. Then, from the initial and final distances between different initial conditions, the local rate of separation is calculated.

**LFNs: a network description of flow systems.** We consider a class of networks in which nodes represent discrete subregions of given domain while the geometry of the links describes a transport process taking place on it during a precise interval of time. Such networks, in the most general case, are thus directed, weighted, and temporal, and each link weight quantifies the importance of a transport event occurred between a pair of nodes starting from time $t_0$ and for a duration of $\tau$. Local measures computed on single nodes of the network (e.g., degrees, strengths, etc.) are thus expected to highlight transport patterns at different spatio-temporal scales[1,34] characterized by $\tau$ and by the spatial subregion's size. LFNs construction is based on the discretization of a metric space $D$ in a fine partition in boxes, $\{\mathcal{B}_i; i = 1, 2, ..., L\}$, characterized by a linear size $\chi$. This set of boxes is identified uniquely with the nodes of the network. Then, to each pair of nodes $i$ and $j$, a directed link with a weight $\mathbf{A}(t_0, \tau)_{ij}$ is assigned and it corresponds to the amount of volume $m$ present in $\mathcal{B}_i$ at time $t_0$ that is found in $\mathcal{B}_j$ after a time $\tau$:

$$\mathbf{A}(t_0, \tau)_{ij} = m\left( \mathcal{B}_i \cap \Phi_{t_0+\tau}^{-\tau}(\mathcal{B}_j) \right), \tag{13}$$

where $\Phi_{t_0}^{\tau}$ is the time evolution operator from time $t_0$ to $t_0 + \tau$. Numerical estimations of $\mathbf{A}(t_0, \tau)$ can be done by seeding in $\mathcal{B}_i$ a large number of initial conditions, i.e., Lagrangian particles, following their trajectories for a time $\tau$, and counting how many ended into each $\mathcal{B}_j$. We define the network out-degree and in-degree of node $i$, respectively, as:

$$K_i^O(t_0, \tau) = \sum_j \begin{cases} 1 & \text{if } \mathbf{A}(t_0, \tau)_{ij} > 0 \\ 0 & \text{otherwise} \end{cases}, $$
$$K_i^I(t_0, \tau) = \sum_j \begin{cases} 1 & \text{if } \mathbf{A}(t_0, \tau)_{ji} > 0 \\ 0 & \text{otherwise} \end{cases}. \tag{14}$$

Similarly we also define the out-strength and in-strength of node $i$ as:

$$S_i^O(t_0, \tau) = \sum_j \mathbf{A}(t_0, \tau)_{ij},$$
$$S_i^I(t_0, \tau) = \sum_j \mathbf{A}(t_0, \tau)_{ji}. \qquad (15)$$

Using Eq. (15), two normalizations for the matrix $\mathbf{A}(t_0, \tau)_{ij}$ can be defined

$$\mathbf{P}^f(t_0, \tau)_{ij} = \frac{\mathbf{A}(t_0, \tau)_{ij}}{S_i^O(t_0, \tau)},$$
$$\mathbf{P}^b(t_0, \tau)_{ij} = \frac{\mathbf{A}(t_0, \tau)_{ij}}{S_j^I(t_0, \tau)}. \qquad (16)$$

Since $\mathbf{A}(t_0, \tau)_{ij} \geq 0$, $\mathbf{P}^f(t_0, \tau)_{ij}$ is a row-stochastic matrix while $\mathbf{P}^b(t_0, \tau)_{ij}$ is column-stochastic. Hence, $\mathbf{P}^f(t_0, \tau)_{ij}$ can be interpreted as the probability for a Lagrangian particle to reach the box $\mathcal{B}_j$ at time $t_0 + \tau$, under the condition that it started from a uniformly random position within box $\mathcal{B}_i$ at time $t_0$. Analogously, $\mathbf{P}^b(t_0, \tau)_{ij}$ corresponds to the probability for a particle of having started from $\mathcal{B}_i$ at time $t_0$, under the condition of being found in a random position within $\mathcal{B}_j$ at time $t_0 + \tau$. Thus, $\mathbf{P}^f(t_0, \tau)_{ij}$ is also the forward-in-time probability for a random walker to jump from node $i$ at $t_0$ to $j$ in a time $\tau$ while $\mathbf{P}^b(t_0, \tau)_{ij}$ is the backward-in-time probability to go from $j$ to $i$. A relationship between the degrees and the FTLEs defined above was found in ref. [34]. The degree of a node turns out to be given, to a good approximation, to an average of the stretching factor (12) over the initial conditions contained in the node

$$K_i^O(t_0, \tau) \approx \frac{1}{m(\mathcal{B}_i)} \int_{\mathcal{B}_i} d\mathbf{x}_0 e^{\tau \lambda(\mathbf{x}_0, t_0, \tau)},$$
$$K_i^I(t_0, \tau) \approx \frac{1}{m(\mathcal{B}_i)} \int_{\mathcal{B}_i} d\mathbf{x}_0 e^{\tau \lambda(\mathbf{x}_0, t_0 + \tau, -\tau)}. \qquad (17)$$

These relationships are used, in the limit of sufficiently small nodes, to derive Eq. (4).

**Approximated analytical solutions.** Even though in the rest of the paper, we always use the formulation of Eq. (5), making some assumptions we can evaluate the integral of Eq. (5) to obtain approximate expressions for $B^L$. Indeed, if we assume the stretching dynamics to be purely exponential with almost constant rates $c_f$, $c_b$, we can write

$$\frac{||\delta\mathbf{x}_i(t, \tau)||_{\max}}{||\delta\bar{\mathbf{x}}_i(t)||} \simeq \begin{cases} e^{c_f \tau} & \text{for } T > 0 \\ e^{c_b \tau} & \text{for } T < 0 \end{cases},$$

and, consequently, we would have $\lambda(\mathbf{x}_i, t, T) \simeq c_f$ for $T > 0$ and $\lambda(\mathbf{x}_i, t, T) \simeq c_b$ for $T < 0$. Under this assumption, we can evaluate Eq. (5) as:

$$B_i^L(0, \tau) \simeq \frac{e^{c_b \tau} - e^{c_f \tau}}{\tau(c_b - c_f)}. \qquad (18)$$

If $c_f = c_b$, as appropriate, for instance, for incompressible two-dimensional flows, using the l'Hôpital rule in Eq. (18), we find:

$$B_i^L(0, \tau) \simeq e^{c\tau}, \qquad (19)$$

where we defined $c = c_f = c_b$. This shows that, under the stated conditions, the Lagrangian betweenness $B_i^L(0, \tau)$ increases with the transport time $\tau$ and with the intensity of stretching occurring in node $i$.

**Numerical evaluation of Lagrangian betweenness.** In order to numerically evaluate $B^L$, a discretized version of Eq. (5) is necessary. It can be written as:

$$B_i^L(0, \tau) = \frac{1}{\tau} \sum_{\alpha=0}^N e^{t_\alpha \lambda(\mathbf{x}_i, t_\alpha, -t_\alpha)} e^{(\tau - t_\alpha)\lambda(\mathbf{x}_i, t_\alpha, \tau - t_\alpha)} \Delta t_\alpha, \qquad (20)$$

where $\alpha$ is the discrete index labeling contiguous intermediates times $t_\alpha$ and $N$ is the total number of time steps of durations $\Delta t_\alpha$ used in the discretization of the integrals.

**Comparing Lagrangian betweenness and most probable paths (MPPs) betweenness.** We now compare Lagrangian betweenness with other betweenness centrality definitions that have been introduced and used [10,47] in the context of LFNs. These last quantities are based on a multistep description of the dynamics. Hence, we denote a generic path $\mu$ of $M$-steps between nodes $i$ and $j$ as a $(M + 1)$-uplet $\mu \equiv \{i, k_1, \ldots, k_{M-1}, j\}$ providing a sequence of nodes crossed to reach $j$ at time $t_M$ from $i$ at time $t_0$. Assuming a Markovian dynamics, the forward-in-time probability for a random walker to take the path $\mu$ under the condition of starting at $i$ is [9,10,11,47]

$$\left( p_{ij}^f \right)_\mu = \mathbf{P}_{ik_1}^{f(1)} \left[ \prod_{l=2}^{M-1} \mathbf{P}_{k_{l-1}k_l}^{f(l)} \right] \mathbf{P}_{k_{M-1}j}^{f(M)}, \qquad (21)$$

where $\mathbf{P}^{f(l)}$ corresponds to $\mathbf{P}^f(t_{l-1}, \Delta t)$ with $t_l = (t_0 + l\Delta t)$ and $l = \{1,\ldots,M\}$ ("Methods"). Hence, $\Delta t$ is the duration of a single step and is assumed to be constant in the whole path. Consequently, the backward-in-time probability to take the path $\mu$ under the condition of starting at $j$ is

$$\left( p_{ij}^b \right)_\mu = \mathbf{P}_{ik_1}^{b(1)} \left[ \prod_{l=2}^{M-1} \mathbf{P}_{k_{l-1}k_l}^{b(l)} \right] \mathbf{P}_{k_{M-1}j}^{b(M)}, \qquad (22)$$

where $\mathbf{P}^{b(l)}$ corresponds to $\mathbf{P}^b(t_{l-1}, \Delta t)$ with $t_l = (t_0 + l\Delta t)$ and $l = \{1,\ldots,M\}$. Maximizing Eqs. (21) and (22) over the intermediate nodes, we are able to find the MPP connecting each pair of nodes $i$, $j$ forward and backward in time, respectively. With the whole set of MPPs at hand, we can now provide a probability-based definition of betweenness centrality. We define the forward- and backward-in-time MPP-betweenness at $M$-steps as:

$$B_i^{fMPP} = \sum_{l,m} g_i(l; m), \qquad (23)$$

$$B_i^{bMPP} = \sum_{l,m} h_i(l; m), \qquad (24)$$

where $g_i(l; m) = 1$ or $h_i(l; m) = 1$ when $i$ is crossed by the forward or backward MPP between $l$ and $m$, respectively, and zero otherwise. Then, we can finally introduce the symmetrized-in-time MPP-betweenness of Eq. (25) as an average of $B_i^{fMPP}$ and $B_i^{bMPP}$

$$\bar{B}_i^{MPP} = \sum_{l,m} g_i(l; m) + \sum_{l,m} h_i(l; m). \qquad (25)$$

We can now compare the Lagrangian betweenness calculated from Eq. (20) and the betweenness explicitly calculated from MPPs in LFNs from Eq. (25). Note that any network-based formulations of betweenness implies inherently a discrete description of the dynamics since network paths are discontinuously composed by different steps. As such, to perform properly the aforementioned comparison, it is necessary to match the temporal discretization scales of $B^L$ and $\bar{B}^{MPP}$ by setting the number of time steps $N$ of Eq. (20) equal to the number of steps $M$ used for the calculation of $\bar{B}^{MPP}$. Supplementary Fig. 1 illustrates, for the double-gyre flow, $B^L$ and $\bar{B}^{MPP}$ fields for $N = M = 2, 3, 5$ in the [0, 15] time interval: high-betweenness regions are organized in narrow lines that, for a given $N = M$, create identical spatial patterns both for $B^L$ and $\bar{B}^{MPP}$. While the characteristic values of $B^L$ do not depend on the number of steps, $\bar{B}^{MPP}$ values increase for larger $M$, becoming noisier. Possibly, such discrepancy is related to two main factors: (i) $\bar{B}^{MPP}$ uses only the MPPs while $B^L$ accounts for all paths, (ii) the added numerical diffusion due to the discretization of space, which is less important in the two-step paths used for $B^L$ because of the smaller number of steps. To quantitatively investigate the similarity among $B^L$ and $\bar{B}^{MPP}$ patterns, since the spatial resolution of $B^L$ is higher than the one of $\bar{B}^{MPP}$, we averaged the values of $B^L$ inside each network node and we compared such averages with the corresponding values of $\bar{B}^{MPP}$. The resulting Spearman correlation coefficients are: 0.90, 0.90, 0.86 for $N = M = 2, 3, 5$, respectively, confirming the strong agreement between both quantities.

**Theoretical and oceanic flow fields for numerical evaluations.** The double gyre[29] is a two-dimensional time-periodic flow defined in the rectangular region of the plane $\mathbf{x} = (x, y) \in [0; 2] \times [0; 1]$. It is characterized by the stream function

$$\psi(x, y, t) = A \sin(\pi f(x, t)) \sin(\pi y), \qquad (26)$$

with

$$f(x, t) = a(t)x^2 + b(t)x, \qquad (27)$$

$$a(t) = \gamma \sin(\omega t), \qquad (28)$$

$$b(t) = 1 - 2\gamma \sin(\omega t). \qquad (29)$$

From these expressions, the two components of the velocity are

$$\dot{x} = -\frac{\partial \psi}{\partial y} = -\pi A \sin(\pi f(x, t)) \cos(\pi y), \qquad (30)$$

$$\dot{y} = \frac{\partial \psi}{\partial x} = \pi A \cos(\pi f(x, t)) \sin(\pi y) \frac{\partial f(x, t)}{\partial x}. \qquad (31)$$

Depending on the value taken by the parameter $\gamma$, this theoretical flow field displays different dynamical behaviors, yet sufficiently simple to carefully analyze the underlying structures. For $\gamma = 0$, the flow is steady and fluid particles follow very simple trajectories, rotating along closed streamlines, clockwise in the left half of the rectangular domain, and counterclockwise in its right half. The central streamline $x = 1$, a heteroclinic connection between the hyperbolic point at (1, 1) and the one at (1, 0), acts as a separatrix between the two regions. However, when $\gamma > 0$, more complex behavior, including chaotic trajectories, arises. The periodic perturbation breaks the separatrix, so that some exchange of fluid is possible between the left and the right sub-domains. As parameters, following ref. [29], we chose $A = 0.1$, $\omega = 2\pi/10$, $\gamma = 0.25$ and we focus our analysis on the time interval

[0, 15] setting $t_0 = 0$ and $\tau = 15$. For the calculation of $B^L$, we fill the whole domain with 78804 Lagrangian particles regularly spaced and we reconstruct each trajectory using a Runge–Kutta 4th-order integration algorithm with temporal step of 0.05. For the calculation of $\bar{B}^{MPP}$, we use instead 2001,000 particles uniformly seeded in 20,000 square boxes representing network nodes and the same Runge–Kutta 4th-order integration scheme.

For the Adriatic Sea application, we use the horizontal near-surface currents simulated by a data-assimilative operational ocean product at $(1/16)°$ of resolution over the Mediterranean basin, provided by E.U. Copernicus Marine Environment Service Information website (https://doi.org/10.25423/medsea_reanalysis_phys_006_004). Further information on this model can be found in refs. [65]. Among the 72 horizontal layers resolved by the model, we focus on surface ocean dynamics by seeding Lagrangian particles on a regular grid of $(1/20)°$ of resolution over the 15 m depth layer and considering only the horizontal velocity. Particles were advected with a 4th-order Runge–Kutta scheme with a time step of 3 h to generate trajectories that were then used to compute the FTLEs and $B^L$, which was calculated according to Eq. (20), by considering $\Delta t_\alpha = 1$ day. Second, we exploit a gridded velocity field at $(1/8)°$ spatial resolution representing surface geostrophic currents computed from remote-sensed SSH. Altimetric products (SSH, sea level anomaly, and the 20-year mean geoid) come from the regional SSALTO/DUACS gridded multi-mission altimeter product, processed by SSALTO/DUACS and distributed by Aviso+ (https://www.aviso.altimetry.fr). This horizontal velocity field was used to compute $B^L$ as explained before, while seeding particles over a regular grid of resolution of $(1/40)°$.

The horizontal velocity fields used for the Kerguelen region come from the Kerguelen altimetry regional product. This product, specifically calibrated for the region, was also processed by SSALTO/DUACS and distributed by Aviso+ (https://www.aviso.altimetry.fr). The velocity field possesses a $(1/8)°$ spatial resolution and were used to compute $B^L$ with the same scheme illustrated for the Mediterranean Sea, with a resolution of $(1/40)°$.

The velocity field used in the north Pacific region is the GLORYS12V1 product (`GLOBAL_REANALYSIS_PHY_001_030`) provided by E.U. Copernicus Marine Environment Service Information website. This global reanalysis product combines track altimeter data, satellite sea-surface temperature, sea ice concentration, in situ temperature, and salinity vertical profiles. It has a spatial resolution of 1/12°, a temporal resolution of 1 day, and 50 vertical layers. Particles were seeded on a regular grid of $(1/20)°$ of resolution over the 15 m depth layer and advected with a 4th-order Runge–Kutta scheme with a time step of 3 h.

**Plankton abundance data and evenness calculation**. The plankton data presented here were collected during a cruise to the Kursohio Extension Front in October 2009. The Kuroshio Extension is the eastward flowing extension of the Kuroshio western boundary current that is deflected off the coast of Japan. The Kuroshio Extension is characterized by strong zonal velocities and strong mesoscale and submesoscale dynamics. The goal of the cruise was to undertake a high spatial resolution physical–chemical–biological survey of the front to explore the connections between physical frontal processes and the plankton community at fine scales. The sampling strategy for the chemical and biological data collected during this cruise is fully described in ref. [43], but here we briefly describe the data collection and samples used in this paper. Five ship transects crossed the Kuroshio Extension Front, with 7–8 sampling stations spaced roughly 9 km apart along each transect. At each sampling station, water was collected from the surface using a clean bucket and from five additional depths using Niskin bottles mounted to the CTD rosette. From each of these surface and depth samples, 500 ml were collected and fixed with formaldehyde and then later used for microscopic enumeration of phytoplankton. The microscopy data are openly available from the Pangaea database[66]. From the surface samples, 1000 ml were collected for subsequent qPCR analysis of the abundance of Ostreococcus clades (method fully described in ref. [44]).

We locally estimate plankton diversity at each sampling site by calculating species evenness[45]. For the microscopy dataset, we focused only on surface samples and we identify species with different plankton types while for the Otreococcus dataset, we consider the two clades as two species. In both cases, we refer to the relative abundance of the species $i$ as $p_i$. Thus, for the sampling location $j$, we calculate the species evenness $E_j$ as:

$$E_j = -\frac{\sum_i p_i \log(p_i)}{\log(S_j)} \tag{32}$$

where $S_j$ is the number of species in the location $j$. Note that very similar results are obtained using instead the Shannon index (i.e., the non-normalized evenness). However, we prefer evenness since it provides an absolute and normalized diversity measure that facilitate the comparison between different datasets.

## Data availability

The data-assimilation product used for the Adriatic Sea is available at https://doi.org/10.25423/MEDSEA_REANALYSIS_PHYS_006_004. The altimetry product used for the Adriatic Sea and Kerguelen region is available at https://www.aviso.altimetry.fr. The velocity field used in the north Pacific region is the GLORYS12V1 product (`GLOBAL_REANALYSIS_PHY_001_030`) provided by E.U. Copernicus Marine Environment Service Information website. The microscopy data are available at https://doi.org/10.1594/PANGAEA.819110. The sequences data used have been deposited in GenBank (accession nos. KT012724 to KT013052). The codes used to compute Lagrangian betweenness are available online at https://github.com/serjaaa/lagrangian-betweenness.

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

## Acknowledgements

E.S-G. thanks Ricardo Martínez-García for providing relevant feedback on the early versions of the manuscript. M.F. and E.S-G. are very grateful for support from the Simons Foundation: the Simons Collaboration on Computational BIOgeochemical modeling of Marine EcosystemS (CBIOMES #549931 to M.F.). We acknowledge the Ministerio de Ciencia, Innovación y Universidades (MCIU), the Agencia Estatal de Investigación (AEI) and the European Regional Development Funds (ERDF) for its support to the project <MDM-2017-071>, Maria de Maeztu Program for Units of Excellence in R&D.

## Author contributions

E.S-G. conceived and directed the study, developed the theory, and wrote the manuscript. E.S-G. and A.B. designed oceanic applications. E.S-G., M.F., and S.C. designed biological applications. E.S-G., A.B., and R.V. performed numerical simulations and analysis. E.S-G., A.B., V.R., C.L., and E.H-G provided physical and mathematical interpretations. All authors provided feedback and helped in shaping the manuscript.

## Competing interests

The authors declare no competing interests.
