## [Peer Review File · Nature Communications]

REVIEWER COMMENTS

Reviewer #1 (Remarks to the Author):

Review of Lagrangian betweenness: a measure of bottlenecks in dynamical systems with oceanographic examples.

This paper explores a new formulation of the centrality measure called betweenness. This measure has been in existence for many years within the context of graph theory. This paper adopts an alternative and novel definition of this centrality measure that opens up the applications that are particularly useful to dynamic systems. I support the publication of this paper since the approach will be informative to many disciplines. The paper is well written and the figures suitable. The videos worked to highlight the complexity of the model. There is enough detail to adapt the formulation to a dynamic modelling algorithm.

Some minor comments:

1. Full stop missing at line 65
2. Please justify the dates of the satellite altimetry (line 213).
3. My primary concern is the issue of connection weights or strengths on the model performance. The examples shown have a particular distribution of edge weights but how does the model perform when the distribution is flat or skewed. The degree distribution may be linked to the weight thresholds (if in existence) and hence influence the model significantly. Are there situations where a 'weak' oceanographic system is not robust enough for modelling with this method?
4. There is a need to highlight the improvement at least in computing speed of this new formulation compared to the older network metrics. If this is not practical then some general comments on expected processing gains in oceanographic modelling would be useful.

Reviewer #2 (Remarks to the Author):

In the introduction, the authors write,

"Bottlenecks can then be related to regions where trajectories converge from disparate origins and are scattered away towards several destinations afterwards. ...

The bottleneck notion could in principle be generalized beyond the context of network theory."

Indeed, the bottleneck notion has already been known in dynamical systems theory for the better part of a century, and the authors seem not to be aware of this.

Even in his classic textbook Strogatz (1994) introduces undergrads to the idea of a bottleneck in a dynamical system. Ott (1993) in his textbook (the earlier version of their reference [13]) refers to the same phenomena in terms of tunnels as well as intermittency, where a trajectory spends a long time in large sets and occasionally passes to other large sets via a bottleneck. Chemists have known since the 1930s about state space bottlenecks near hyperbolic points in connection to chemical reactions which

must pass through transition states on the way from reactants to products.

If the goal of the paper is to merge some concepts of network theory with dynamical systems, this failure to appreciate that they have already walked where dynamicists have tread is not a good start.

Even if we forgive this, the authors go on to say, "there is not an analogous measure to quantitatively characterize bottlenecks in dynamical systems as betweenness centrality does in networks". I suggest they look at the nearly century-old transition state theory and quantitative methods toward the identification of a surface of minimal flux. Or consider their reference [13] or Strogatz on the identification of regions of tunnels and bottleneck regions. It is not at all clear that there is a "gap" needed to be filled that needs another mathematical expression.

The authors have a new way to identify important hyperbolic regions in time-dependent dynamical systems, where hyperbolic points and heteroclinic/homoclinic connections are difficult to identify. However Wiggins, in refs [22] and [36] and elsewhere seem to identify hyperbolic points (distinguished hyperbolic trajectories) routinely in time-dependent oceanic flows, using the exponential dichotomy, M-function, and other computational approaches.

Even the use of transition matrices and graph-based methods to discover structure in ocean models is not new. Dellnitz, Froyland and co-workers have done this for more than a decade (e.g., Dellnitz et al (2009) *Nonlin. Processes Geophys.*). Furthermore, Bollt and others have explicitly referred to betweenness and congestion as identifying bottlenecks in network models of dynamical systems.

The authors consider the example of the double gyre and two 2D oceanic flows. They write, "The possibility of detecting transport bottlenecks created by currents at the ocean surface, could be not only relevant for oceanography by itself but also for several applications ranging from pollutants spreading to marine ecology or fishery science." To justify publication in *Nature Communications*, especially in light of the previous comments on the relative lack of novelty of bottlenecks in fluid flows and more generally dynamical systems, as well as the lack of novelty of network/graph-based methods, I believe the authors need to demonstrate their results for a specific application, e.g., comparing with observations of pollutants or algae, or along the lines of the early-warning signals they mention in the discussion.

To speak of the paper's strengths, the authors are very careful to show the robustness of their results, particularly for the Adriatic example. I share with them that the persistence over time of the high betweenness regions in their two ocean examples is both surprising and interesting. To make that claim stronger, though, it would be useful to see depicted that this isn't just a persistent stagnation point from an Eulerian point of view. That is, show how variable the velocity field is. The ocean examples in this sense are simpler than the double gyre, where the gyre hyperbolic point moves across the domain at least once per period, despite the velocity field being relatively simpler. Are the authors picking special cases which show this persistence or is it to be expected as a general feature in ocean flows? In the introduction and discussion, the authors make a good case for a new definition of betweenness and its use in many areas, including, and starting with, geophysical flows.

All that said, I am still left wondering what is of significance in this paper to justify publication in Nat. Comm. There are far more connections between dynamical systems theory and network theory than the two references mentioned [1] and [13]. The authors failure to put their work in proper historical perspective is troubling to say the least. Why should this go to the larger scientific community rather than an oceanography specialized journal?

MINOR POINTS

Given the symmetry of the double gyre, why does the Lagrangian Betweenness ridge break that symmetry? (the semi-circle opens to one side in Fig. 4) Is this an artifact of the times chosen? If other times were chosen, would the semicircle open to the other side?

Furthermore, the authors single out the semicircle ridge, but what about the 'shorter' ridge in the left gyre, with x between 0.5 and 0.7, y between 0.3 and 0.7? Does that region have significance? What's the criteria for choosing only the semicircle?

Reviewer #3 (Remarks to the Author):

Review of "Lagrangian betweenness: a measure of bottlenecks in dynamical systems with oceanographic examples" by Ser-Giacomi et al

General Comment

The authors propose a new methodology to isolate bottlenecks in dynamical systems with applications to Lagrangian transport, extending the notion of betweenness of network theory. They base their approach on the computation of the Finite-Time Lyapunov Exponents (FTLE). They derive a formula (Eq. 5) to measure the Lagrangian Betweenness. This measure is applied in the context of the Lagrangian transport in the theoretical two dimensional double gyre system, and then applied to the two dimensional transport in the ocean. The new measure allows to isolate bottleneck regions, and at the same time provides information on the underlying topological structure of attracting sets. This manuscript provides a very useful new tool worth communicating and promoting to a large scientific community. The methodology is sound and the method is well described and illustrated. The implementation of the method is also provided in details.

I provide here some specific comments that could help to improve the quality of the manuscript.

Specific comments

- The FTLE. As scientist involved in atmospheric and climate dynamics, I was surprised to see the FTLE defined based on the maximum finite time amplification. In a large community, the quantity defined that way (Eqs. 9-10) is the logarithm of the singular value associated with the singular vectors of a symmetric matrix describing the finite-time amplification of small perturbations (Kalnay, 2003,

Atmospheric Modeling, Data Assimilation and Predictability, Cambridge University Press ; Diaconescu & Laprise, 2012, doi.org/10.1016/j.earscirev.2012.05.005). The FTLE in this community is the amplification during a certain interval of time along the Lyapunov vectors (Trevisan and Pancotti, 1998, [https://doi.org/10.1175/1520-0469\(1998\)055<0390:POLVAS>2.0.CO;2](https://doi.org/10.1175/1520-0469(1998)055<0390:POLVAS>2.0.CO;2); Vannitsem, 2017, <https://doi.org/10.1063/1.4979042>). I realize that this is maybe not the case in Lagrangian transport, but it is maybe useful to draw a warning on this.

- The analysis is done in two dimensions and works very well as the measure is defined in restricted 2D state space (equivalent to the physical space). The authors mentioned the potential applications to other dynamical systems. Usually in environmental applications, the state space is much higher. I suspect that the use of this measure in high dimension problems will not provide such clearcut diagnostics as multiple directions of (in)stabilities will be present. I think that it would be worth discussing this question in the manuscript.

- Related to the previous question, did the authors try to apply these techniques to low-order or reduced-order dynamical systems (more than 2 dimensions)? It would be useful to comment on that and maybe show some examples.

- Section VB describes the Lagrangian Betweenness to the Most Probable Path Betweenness. I am wondering whether this comparison is worth putting in the manuscript, as it is not used anywhere, unless I missed something.

RESPONSE LETTER corresponding to manuscript:

“Lagrangian betweenness: a measure of bottlenecks in dynamical systems with oceanographic examples”

MAIN CHANGES

A) Application to plankton ecology

- Following the suggestions of the editor and referee 2, we added results from a new analysis based on two independent plankton abundance datasets from in-situ sampling across the Kuroshio Front, in the Pacific Ocean.
- We examined the relationship of betweenness and diversity in two independent datasets collected during a cruise in October 2009; microscopy-based abundances of plankton types and phylogenetically-derived abundances of *Ostreococcus* clades.
- We used a high-resolution reanalysis ocean product to compute Lagrangian betweenness in the same region and period of the cruise and we measured plankton diversity in terms of species evenness.
- We find remarkable statistically significant correlations between plankton evenness and Lagrangian betweenness, for both datasets.
- This additional study supports that further investigation of ecological bottlenecks, as revealed by high values of Lagrangian betweenness, will be valuable and promising.

B) Novelty and relevance of the theoretical results

- We stressed in the Introduction, Results and Discussions that our main theoretical result is more than a novel method to detect new kinds of bottlenecks in dynamical systems. Indeed, it is the identification of a link between a widely used network measure (betweenness) and dispersion and mixing features (associated to Lyapunov exponents).
- We pointed-out that the nature of the dynamical bottlenecks that we characterize is qualitatively and quantitatively different respect to the ones previously studied in the literature reported by Referee 2.
- We clarified that the aforementioned theoretical connection is of general interest, beyond the field of dynamical systems. It indeed provides new interpretations and numerical implementations for the definition and calculation of betweenness centrality in the field of complex networks.

C) Robustness and applicability

- We added a paragraph in the Discussions to stress: (i) the huge improvement in numerical efficiency given by the use of Lagrangian betweenness respect to the explicit network-based betweenness and (ii) the fact that avoiding the network construction prevents any possible sensitivity issue related to time discretization and weights distribution.

- We added a paragraph in the Discussion to mention the possible challenges related to our approach when applied to high-dimensional dynamical systems.
- We added a paragraph in Section III.B.1 stressing that the persistence and robustness of the Adriatic betweenness peak is not associated to the presence of any Eulerian stagnation point. This highlights the purely Lagrangian origin of such pattern that is not deducible from simpler analysis.

D) Framing and accessibility for a wide readership

- We made clearer and accessible the definition and interpretations of Finite Time Lyapunov Exponents (FTLEs) for a wider readership adding some references suggested by referee 3 (in the Introduction and Methods) as well as expanding the corresponding Section in the Methods.
- We revised the Introduction, Results and Discussions to better frame our theoretical results with respect to the existing literature and we added references suggested by referee 2.
- We added a sentence in the Acknowledgements to include the link (not active yet) of the open-source repository used to compute Lagrangian Betweenness.

POINT-BY-POINT RESPONSE TO REVIEWER 1

(Referee's comments are reported in *italic* while our answers immediately follow in **red font**. The brackets { } specify the lines that have been edited in the manuscript.)

This paper explores a new formulation of the centrality measure called betweenness. This measure has been in existence for many years within the context of graph theory. This paper adopts an alternative and novel definition of this centrality measure that opens up the applications that are particularly useful to dynamic systems. I support the publication of this paper since the approach will be informative to many disciplines. The paper is well written and the figures suitable. The videos worked to highlight the complexity of the model. There is enough detail to adapt the formulation to a dynamic modelling algorithm.

Some minor comments:

1. Full stop missing at line

65 **Thanks, corrected.**

2. Please justify the dates of the satellite altimetry (line 213).

Being the Adriatic and Kerguelen studies two example applications, we do not have precise constraints on a specific month and year. We have chosen the month of December because it is representative of the average winter/summer conditions in the northern/southern hemisphere. In this way we covered two 'opposite' seasons.

We added a sentence to clarify this at the beginning of the section devoted to the applications to ocean transport {187-189}. Please note that the approach is of course totally different for the new application to plankton ecology: in this case, to match our analysis with the in-situ sampling, we restricted our calculation to the exact days of the cruise.

3. My primary concern is the issue of connection weights or strengths on the model performance. The examples shown have a particular distribution of edge weights but how does the model perform when the distribution is flat or skewed. The degree distribution may be linked to the weight thresholds (if in existence) and hence influence the model significantly. Are there situations where a 'weak' oceanographic system is not robust enough for modelling with this method?

We agree that the network approach could present (as every other method) some sensitivity issues. In addition to the weight distribution, that in our case is not so critical since we focus on degrees and we do not use thresholds, the approximation associated to the time discretization is also worth to mention. The good thing is that, once we directly use Eq. (5) for our calculation, we bypass entirely the network construction process. Thus, this permits to neglect the aforementioned possible issues.

We added a paragraph about this and numerical efficiency in the Discussions {299-302} (see also MAIN CHANGES).

4. There is a need to highlight the improvement at least in computing speed of this new formulation

compared to the older network metrics. If this is not practical then some general comments on expected processing gains in oceanographic modelling would be useful.

We totally agree that a proper in-depth analysis of the numerical performance of our trajectory-based betweenness versus explicit-network measures would be of great interest for future applications. We think that, mainly for space limitations and to avoid technicalities, this does not match the scope of the paper.

Nevertheless, we added a paragraph in the Discussions to discuss in general terms the large improvement in performance of our approach with respect to the network-based computation {299-302} (see also MAIN CHANGES). We also plan to make public our codes, making possible for everybody to test independently their efficiency.

POINT-BY-POINT RESPONSE TO REVIEWER 2

(Referee's comments are reported in *italic* while our answers immediately follow in **red font**. The brackets { } specify the lines that have been edited in the manuscript.)

In the introduction, the authors write, "Bottlenecks can then be related to regions where trajectories converge from disparate origins and are scattered away towards several destinations afterwards. ... The bottleneck notion could in principle be generalized beyond the context of network theory."

We fully agree with the reviewer 2 that our historical perspective of the study was far from being complete, which was mainly due to the space limitation in the journal. In this new version of the manuscript we have followed closely his/her suggestions to try to solve this issue. We have improved our discussion on bottlenecks in dynamical systems and, more generally, network theory.

However, the main theoretical result of this study is not only the introduction of a novel and efficient way to detect bottlenecks and/or hyperbolic regions in dynamical systems, but the discovery (mathematically and numerically supported) of the link between a widely used network measure (betweenness) and Lyapunov exponents. Moreover, exploiting the relationship between FTLE ridges and hyperbolicity another step can be made, finally connecting the measure of betweenness in a network with hyperbolic regions of the associated dynamical system. To our knowledge, there is no work published illustrating such a relationship.

Another fundamental point is that the aforementioned relationship can be fruitfully exploited not only from the dynamical system 'side': we believe that for network scientists it would be of great interest connecting a metric like betweenness with dynamical properties such as predictability, dispersion and mixing typically described with Lyapunov exponents. Indeed, betweenness, up to now, has been defined and interpreted from a solely algorithmic way (i.e. by numerically counting shortest paths).

More generally, even if we assume that two different concepts are well characterized in their respective fields, there is an indubitable scientific interest into discovering a quantitative relation between them, thus contributing to bridging different areas of research. Multidisciplinary approaches like these are exactly in the line with the main scopes of a journal like Nature Communications.

Note finally that, forgetting for a moment about dynamical systems, our alternative definition of betweenness centrality (see Eq. 3), based on a temporal convolution of in- and out-degrees, is completely new and could, by itself, contribute to the network field providing a faster and most comprehensive (since it refers to all the paths, not only the shortest ones) alternative to the classic formulation. Moreover, as highlighted in the discussion, the definition of Eq. (3) poses also a more fundamental question to the network community: to which degree is betweenness centrality an independent property of a network or just a byproduct that is implicitly determined by the degree distribution?

Indeed, the bottleneck notion has already been known in dynamical systems theory for the better part of a century, and the authors seem not to be aware of this.

We agree with the referee that we did not sufficiently discuss how the concept of bottleneck has been previously conceived in dynamical systems. However, please note that our theoretical definition, detection and interpretation of bottlenecks is different from the ones mentioned by the referee. We clarified this in the following and in the revised version of the paper.

The word 'bottleneck' may express related but different concepts in phase space: (i) it may be applied to

regions where trajectories spend a long time; (ii) it can denote (hyper)surfaces crossed by minimal fluxes; (iii) it can refer to regions acting as hubs, i.e., where trajectories converge from many different places and are then scattered to also many different ones. The first two meanings are the most studied in the literature while the last one is the one developed in our work. Our conceptualization of bottlenecks is associated to zones with both strong mixing and dispersion (both forward- and backward-in-time) and can be very relevant, for example, in certain biological processes (see also new application in MAIN CHANGES). We have tried to clarify all this in the new version of the manuscript. See below for more details.

Even in his classic textbook Strogatz (1994) introduces undergrads to the idea of a bottleneck in a dynamical system. Ott (1993) in his textbook (the earlier version of their reference [13]) refers to the same phenomena in terms of tunnels as well as intermittency, where a trajectory spends a long time in large sets and occasionally passes to other large sets via a bottleneck.

We have now included and discussed in the text references about previous definitions of the term 'bottleneck' in the context of dynamical systems (those appearing, for example, in the books by Strogatz or Ott; for the case of transition-state theory in chemistry see the response to the following point).

However, as previously indicated, we have also commented a main difference between these definitions and the one we employ from the perspective of networks: in some dynamical systems literature [Strogatz, Ott] bottlenecks or tunnels refer to regions of phase space where trajectories spend a very long time. Instead, our definition refers to areas of phase space where many trajectories go through, coming from diverse origins and departing also to different destinations, independently of the time spent in the central region, as indicated in the concept of betweenness for a node in a graph.

We added an entire new paragraph with references about this in the Introduction {70-76} (see also MAIN CHANGES).

Chemists have known since the 1930s about state space bottlenecks near hyperbolic points in connection to chemical reactions which must pass through transition states on the way from reactants to products. [...] I suggest they look at the nearly century-old transition state theory and quantitative methods toward the identification of a surface of minimal flux.

Certainly, it is pertinent to mention the concept of surface of minimal flux in the transition-state theory of chemical kinetics and we discussed also this in the revised manuscript. These (hyper)surfaces are said to have the bottleneck property if they separate the regions of the energy surface describing products and reactants with specific properties, namely that the only way a trajectory can represent a transition from reactants to products is to pass through it, without local recrossing.

However, also in this case, although it is likely that in some cases the surfaces of transition-state theory represent bottlenecks in the sense used in the present paper (i.e. hubs), the two concepts are in principle different, as the definition of the minimal surface relies in strictly local topological properties of phase space.

We added an entire new paragraph also about this in the Introduction {70-76} (see also MAIN CHANGES).

"there is not an analogous measure to quantitatively characterize bottlenecks in dynamical systems as betweenness centrality does in networks". [...] It is not at all clear that a "gap" needed to be filled that needs another mathematical expression.

As already explained above, the notion of bottleneck introduced here is coming from the network concept of betweenness centrality. Hence, it is conceptually and mathematically different from the measures presented in the references pointed by the referee. We thus believe in the validity of our statement, while, at the same time, we agree that more historical perspective was needed. For these reasons we substituted that sentence with a new paragraph {70-76}.

The authors have a new way to identify important hyperbolic regions in time-dependent dynamical systems, where hyperbolic points and heteroclinic/homoclinic connections are difficult to identify. However Wiggins, in refs [22] and [36] and elsewhere seem to identify hyperbolic points (distinguished hyperbolic trajectories) routinely in time-dependent oceanic flows, using the exponential dichotomy, M-function, and other computational approaches.

We acknowledge that there are several methods in the literature suitable to locate hyperbolic structures. Note that the scope of the present work does not consist into identifying such structures. Our main objective is locating trajectory hubs, and the fact that they are related to hyperbolic structures is an outcome of our approach.

We added a sentence and citations to the references mentioned by the referee in the revised manuscript {83-84, 142-143} (see also MAIN CHANGES).

Even the use of transition matrices and graph-based methods to discover structure in ocean models is not new. Dellnitz, Froyland and co-workers have done this for more than a decade (e.g., Dellnitz et al (2009) Nonlin. Processes Geophys.). Furthermore, Bollt and others have explicitly referred to betweenness and congestion as identifying bottlenecks in network models of dynamical systems.

It is obvious that the novelty of our work is not in the introduction of the betweenness metric and/or in the use of network approaches to geophysics. There are already several works in the literature on these topics. Hence, the concept of bottleneck in terms of betweenness is the same as the one used by Bollt et al, and the one in our previous work in ref. [9] (where we give proper credit to the work by Bollt et al.) that we summarize in Sect. 5F.

The novelty here, as already pointed out before, is that we relate this network concept to Lyapunov exponents, thus making an explicit link to dynamical systems.

We have now added some sentences in the manuscript citing work by Bollt et al. and, at the same time, we also stressed that betweenness is expressed in terms of Lyapunov Exponents, a measure of stretching along trajectories which is not a priori related to the network framework {421-422} (see also MAIN CHANGES).

I believe the authors need to demonstrate their results for a specific application, e.g., comparing with observations of pollutants or algae, or along the lines of the early-warning signals they mention in the discussion.

We fully agree that showing another application of our approach in addition to the two already presented (focused on realistic physical transport) would add value to the manuscript. Thus, following the editor and referee 2 suggestions, we applied our methodology to the study of local plankton diversity patterns derived by in-situ observations. We found very promising statistical correlations and we added a section to the Results of the paper about this new analysis (see also MAIN CHANGES).

To speak of the paper's strengths, the authors are very careful to show the robustness of their results, particularly for the Adriatic example. I share with them that the persistence over time of the high betweenness regions in their two ocean examples is both surprising and interesting. To make that claim stronger, though, it would be useful to see depicted that this isn't just a persistent stagnation point from an Eulerian point of view. That is, show how variable the velocity field is. The ocean examples in this sense are simpler than the double gyre, where the gyre hyperbolic point moves across the domain at least once per period, despite the velocity field being relatively simpler. Are the authors picking special cases which show this persistence or is it to be expected as a general feature in ocean flows? In the introduction and discussion, the authors make a good case for a new definition of betweenness and its use in many areas, including, and starting with, geophysical flows.

Following the referee suggestion, we further investigated the nature of the betweenness patterns found in the first two oceanic applications presented in the paper:

- In the Kerguelen region, the high-betweenness strip is of course not associated to any stagnation of the velocity field. Indeed, it actually overlaps a region of strong eastward flow that, however, is not stable. This can be easily seen in the Supplementary Video 2 following the movements of Lagrangian tracers.
- For the case of the Adriatic Sea we performed a more in depth analysis to understand the Eulerian features of the circulation. To this aim, we show below two maps (see Figure 1 in this reply) of the average velocity field modulus (from the same high-resolution model) matching the time windows used for the betweenness calculation (i.e. the month of December, $\tau=30$). We first focus only on the 2013 (top panel) and then we also provide a 2002-2013 climatology (bottom panel). We clearly see that the betweenness spot can not be associated to the presence of any stagnating point of the Eulerian velocity field. The average modulus velocity field is indeed smoothed by the temporal variability of the flow during the time window considered. This proves that, despite a marked variability and the absence of clear Eulerian patterns, the high-betweenness region pop-ups driven by a purely Lagrangian dynamics.

We added a paragraph about this in the Results {222-224} (see also MAIN CHANGES).

There are far more connections between dynamical systems theory and network theory than the two references mentioned [1] and [13].

As already state before, we believe that the theoretical novelty of our contribution is in the quantitative link between betweenness and FTLEs. Such direct link is, to our knowledge, new and widely relevant. Indeed, both betweenness and FTLEs are, in their respective fields, key measures to address dynamical features of a large variety of systems.

Why should this go to the larger scientific community rather than an oceanography specialized journal?

Please also refer to our general comment above. Note that, even if we hypothetically assume that our contribution would not be particularly relevant for the dynamical systems field (and we proved that it is not the case), it will bring new concepts and tools in the wider and rapidly growing complex network field.

Moreover, in light of the promising analyses presented in our last application to plankton ecology, the

interest of Lagrangian betweenness will most likely spread across several fields of environmental science (not only oceanography). Indeed, as stressed in the introduction and discussions, our approach is sufficiently general to be applied to any velocity field describing different geophysical flows, definitively not limiting to the ocean.

MINOR POINTS

Given the symmetry of the double gyre, why does the Lagrangian Betweenness ridge break that symmetry? (the semi-circle opens to one side in Fig. 4) Is this an artifact of the times chosen? If other times were chosen, would the semicircle open to the other side?

In this example, as showed in Fig. 5, the ridge of betweenness corresponds to the trajectory of the hyperbolic point in the given time window. Yes, during the successive semi-period, the stable and unstable manifolds will flip to the other side and, consistently, the betweenness ridge will be on the other side as well. Thus, this is not an 'artifact', it just follows from the time evolution of the main manifolds in the time-dependent double-gyre system.

Furthermore, the authors single out the semicircle ridge, but what about the 'shorter' ridge in the left gyre, with x between 0.5 and 0.7, y between 0.3 and 0.7? Does that region have significance? What's the criteria for choosing only the semicircle?

In this system there are, as the referee noted, several, more or less important, ridges of FTLEs. Of course the structure identified by the referee is relevant but it is also weaker than the main ridge (note that the colormap is in logarithmic scale). For length limitation and to avoid too technical discussions, we decided to not discuss other smaller features of the betweenness field in the double-gyre system.

Figure 1

POINT-BY-POINT RESPONSE TO REVIEWER 3

(Referee's comments are reported in *italic* while our answers immediately follow in **red font**. The brackets { } specify the lines that have been edited in the manuscript.)

General Comment

The authors propose a new methodology to isolate bottlenecks in dynamical systems with applications to Lagrangian transport, extending the notion of betweenness of network theory. They base their approach on the computation of the Finite-Time Lyapunov Exponents (FTLE). They derive a formula (Eq. 5) to measure the Lagrangian Betweenness. This measure is applied in the context of the Lagrangian transport in the theoretical two dimensional double gyre system, and then applied to the two dimensional transport in the ocean. The new measure allows to isolate bottleneck regions, and at the same time provides information on the underlying topological structure of attracting sets. This manuscript provides a very useful new tool worth communicating and promoting to a large scientific community. The methodology is sound and the method is well described and illustrated. The implementation of the method is also provided in details.

I provide here some specific comments that could help to improve the quality of the manuscript.

Specific comments

- The FTLE.

As scientist involved in atmospheric and climate dynamics, I was surprised to see the FTLE defined based on the maximum finite time amplification. In a large community, the quantity defined that way (Eqs. 9-10) is the logarithm of the singular value associated with the singular vectors of a symmetric matrix describing the finite- time amplification of small perturbations (Kalnay, 2003, Atmospheric Modeling, Data Assimilation and Predictability, Cambridge University Press ; Diaconescu & Laprise, 2012, doi.org/10.1016/j.earscirev.2012.05.005). The FTLE in this community is the amplification during a certain interval of time along the Lyapunov vectors (Trevisan and Pancotti, 1998, [https://doi.org/10.1175/1520-0469\(1998\)055<0390:POLVAS>2.0.CO;2](https://doi.org/10.1175/1520-0469(1998)055<0390:POLVAS>2.0.CO;2); Vannitsem, 2017, <https://doi.org/10.1063/1.4979042>). I realize that this is maybe not the case in Lagrangian transport, but it is maybe useful to draw a warning on this.

Definition of Eq. (9) of the finite Time Lyapunov exponents is exactly the same as the one mentioned by the reviewer in terms of the singular value of a symmetric matrix. In fact such matrix is the Cauchy-Green tensor we mentioned in the previous version, which is given by M^*M , where M is the Jacobian matrix of the Flow map and M^* its adjoint.

We have clarified this in the revised manuscript adding more explanations in the Methods as well as including the references suggested by the referee {360-376} (see also MAIN CHANGES).

- The analysis is done in two dimensions and works very well as the measure is defined in restricted 2D state space (equivalent to the physical space). The authors mentioned the potential applications to other dynamical systems. Usually in environmental applications, the state space is much higher. I suspect that the use of this measure in high dimension problems will not provide such clearcut

diagnostics as multiple directions of (in)stabilities will be present. I think that it would be worth discussing this question in the manuscript.

We totally agree with the referee that higher-dimensional systems could be challenging. We suppose that, in such kind of applications, Lagrangian betweenness could present the same kind of performance and limitations that the Finite-Time-Lyapunov-Exponents. We also speculate that, for some types of 3-dimensional systems (for instance the ABC flow), since FTLEs are performing well, the Lagrangian Betweenness should also work properly.

We added a new paragraph in the discussions about the aforementioned possible challenges when dealing with high-dimensional systems {303-306} (see also MAIN CHANGES).

- *Related to the previous question, did the authors try to apply these techniques to low-order or reduced-order dynamical systems (more than 2 dimensions)? It would be useful to comment on that and maybe show some examples.*

We also believe that such analysis will be of great interest, however, to respect length limitations and to avoid too technical additions, we decided to not include other applications to the manuscript.

- *Section VB describes the Lagrangian Betweenness to the Most Probable Path Betweenness. I am wondering whether this comparison is worth putting in the manuscript, as it is not used anywhere, unless I missed something.*

We explained how we can calculate betweenness in a network-explicit manner from Most Probable Paths (MPPs) because we use such formulation in Section III.A to prove numerically that the Lagrangian Betweenness patterns in the double-gyre match very well the ones obtained (at very high computational cost) from MPP-betweenness. We report in the text the correlation coefficients and the plots in Supplementary Fig. 1. That is why we believe that a concise introduction of MPP-betweenness in the Methods is useful.

REVIEWER COMMENTS

Reviewer #1 (Remarks to the Author):

Review of Lagrangian betweenness: a measure of bottlenecks in dynamical systems with oceanographic examples.
Revision 2

I thank the authors for careful consideration of the points raised in the first review. However my primary concern regarding the influence of the connection weights or strengths on the model performance remains unresolved.

The model presented does indeed not require a formulation for weighting but this assumes the foundation model (Sea height often) has been through this process. This is of concern since the robustness of the model as shown is predicated on this threshold value without any sensitivity control. Even in Supplementary Fig 5 and examples the weighting of the edges is discussed but without consequence. There is a risk that given the degree values being adopted that the shape of the degree distribution is highly influential on the results.

The models need to be run across a small range of thresholds (i.e Adriatic Sea 20cm) to examine the relationship of degree distribution and oceanographic model strength. Again, are there situations where a 'weak' oceanographic system is not robust enough for modelling with this method?

The rest of the manuscript is shaping up well.

Reviewer #2 (Remarks to the Author):

I think the authors did a good job addressing my concerns, many as they were, and I believe this will lead to a more agreeable reception of the paper. The key thing they have convinced me of is the novelty of the quantitative link between betweenness and Lyapunov exponents, which is very interesting. I still think that betweenness as related to identifying a trajectory hub is very much related to transition states, not in the sense of a minimal surface, but more specifically, the isolating blocks near high dimensional saddle points which are indeed trajectory hubs for trajectories on the way from one large region of phase space to another. But they sort of convince me by stating in their rebuttal that such situations are related to "strictly local topological properties of phase space". That's correct, the 'hubs' in the cases I have in mind are related to topology of the autonomous phase space. And they, by contrast, are identifying regions in a non-autonomous system, where there is no topological property (perhaps there is in the PDE phase space that generates the fluid, but that's another story).

In short, good paper, and I echo the hope that it will be used by practitioners in environmental flows.

Reviewer #3 (Remarks to the Author):

The authors have appropriately addressed the comments I made on the first version. I recommend publication of this manuscript.

RESPONSE LETTER corresponding to manuscript:

“Lagrangian betweenness: a measure of bottlenecks in dynamical systems with oceanographic examples”

We greatly acknowledge the very positive reports of the reviewers. Reviewers #2 and #3 recommend our paper for publication as it is. Concerning reviewer #1 he/she thanks that most of the points raised were properly addressed, but singles out an aspect, related to the robustness of the model.

We interpret that the reviewer refers to the sensitivity of our study (that in general terms involve the analysis of Lagrangian flow networks, Lyapunov exponents, and the Betweenness centrality measure) to the outputs of the different oceanic models that we use for the realistic applications of our study. In fact, this is an important point that has been addressed in previous publications and that we now reference and comment in the new version of the manuscript:

- Concerning the use of Lagrangian Flow Network (LFN) study in the oceanic applications, an exhaustive assessment of sensitivity and robustness of network-derived metrics against the most relevant parameters used in their construction (node size, density of released particles, threshold for considering nodes connections, and integration time) was performed in Monroy et al (2017) *ICES Journal of Marine Science* 74, 1763–1779, <https://doi.org/10.1093/icesjms/fsw235>. The general conclusion of this careful study was that the LFN and the metrics that are obtained from them are robust against a large range of the values of these parameters.

- However, in our methodology we completely bypass the network construction approach by replacing it by expressions of betweenness in terms of Lyapunov exponents. Thus, the pertinent sensitivity analysis is related to the reliability of Lyapunov exponents in the oceanic surface when facing real data. Notably, this was also addressed in Hernandez-Carrasco et al (2011) *Ocean Modelling* 36, 208-218, <http://dx.doi.org/10.1016/j.ocemod.2010.12.006> . In this work the authors address the robustness of Lyapunov exponents against the two most important issues affecting of realistic ocean models: lack of resolution of small scales, and errors in velocity-field estimations (arising either from oceanographic model imperfections or from observational errors). The authors show that, even when some dynamics is missed in the computation of the Lyapunov, the shape and topology of the oceanic structures, which is what matters for the betweenness computation, is still accurate. Thus, since betweenness is calculated from the Lyapunov exponents, this study widely supports the robustness and reliability of our calculations when facing realistic ocean data.

We have added some lines in the Discussions (sect. IV.B) with these comments besides adding the references by Monroy et al (2017) and Hernandez-Carrasco et al. (2011) to the bibliography.

We hope that with these changes the manuscript would be found appropriate for publication.

Kind regards,

Dr. Enrico Ser-Giacomi, on behalf of all authors

REVIEWERS' COMMENTS

Reviewer #1 (Remarks to the Author):

I thank the authors for consideration of the issues raised and extra effort required to advance the paper. I am prepared to accept the changes as being sufficient for publication. However I do note that future studies with this very promising method do need to fully consider the statistical aspects of robustness. In my own work which is more ecologically based the links are randomly subset (bagging) to determine the contribution of links to centrality measures. I also explore random switching of link end nodes whilst maintaining degree distribution. I combine these changes while also using repeated variations in thresholds to determine the model response. Often there are 'sweet spots' that are worth exploring in the model response especially if the topology resembles small world or scale free systems. These comments aside I look forward to the publication of this paper.